



# MESMER-M: an Earth System Model emulator for spatially resolved monthly temperatures

Shruti Nath[1,2], Quentin Lejeune[1], Lea Beusch[2], Carl-Friedrich Schleussner[1,3], and Sonia I. Seneviratne[2]

[1]Climate Analytics, Berlin, Germany
[2]Institute of Atmospheric and Climate Science, ETH Zurich, Zurich, Switzerland
[3]Integrative Research Institute on Transformations of Human-Environment Systems (IRI THESys) and Geography Department, Humboldt-Universität zu Berlin, Berlin, Germany

**Correspondence:** shruti.nath@climateanalytics.org

**Abstract.** The degree of trust placed in climate model projections is commensurate to how well their uncertainty can be quantified, particularly at timescales relevant to climate policy makers. On interannual to decadal timescales, model uncertainty due to internal variability dominates and is imperative to understanding near-term and seasonal climate events, but hard to quantify owing to the computational constraints on producing large ensembles. To this extent, emulators are valuable tools for approx-
imating climate model runs, allowing for exploration of the model uncertainty space surrounding select climate variables at a significantly reduced computational cost. Most emulators, however, operate at annual to seasonal timescales, leaving out monthly information that may be essential to assessing climate impacts. This study extends the framework of an existing spatially resolved, annual-scale Earth System Model (ESM) emulator (MESMER, Beusch et al. (2020)) by a monthly downscaling module (MESMER-M), thus providing local monthly temperatures from local yearly temperatures. We first linearly represent
the mean response of the monthly temperature cycle to yearly temperatures using a simple harmonic model, thus maintaining month to month correlations and capturing changes in intra-annual variability. We then construct a month-specific local variability module which generates spatio-temporally correlated residuals with month and yearly temperature dependent skewness incorporated within. The performance of the resulting emulator is demonstrated on 38 different ESMs from the 6th phase of the Coupled Model Intercomparison Project (CMIP6). The emulator is furthermore benchmarked using a simple Gradient
Boosting Regressor based, physical model trained on biophysical information. We find that while regional-scale, biophysical feedbacks may induce non-uniformities in the yearly to monthly temperature downscaling relationship, statistical emulation of regional effects shows comparable skill to approaches with physical representation. Thus, MESMER-M is able to generate ESM-like, large initial-condition ensembles of spatially explicit monthly temperature fields, thereby providing monthly temperature probability distributions which are of critical value to impact assessments.

# 1   Introduction

Climate model emulators are computationally cheap devices that derive simplified statistical relationships from existing climate model runs to then approximate model runs that have not been generated yet. By reproducing runs from deterministic, process-based climate models at a significantly reduced computational time, climate model emulators facilitate exploration of the





uncertainty space surrounding model representation of climate responses to specific forcings. A wide toolset of Earth System

Model (ESM) emulators exists with capabilities ranging from investigating the effects of greenhouse gas emission scenarios on global to regional mean annual climate fields (Meinshausen et al., 2011) to looking at regional-scale annual, seasonal and monthly internal climate variabilities (Link et al., 2019; McKinnon and Deser, 2018; Alexeeff et al., 2018; Castruccio et al., 2019).

Recently, a Modular Earth System Model Emulator with spatially Resolved output (MESMER) (Beusch et al., 2020) has

been developed with the ability to represent model uncertainty due to natural climate variability. It does so using a combination of pattern scaling and a natural climate variability module, to generate spatially resolved, yearly temperature realisations that emulate the properties of ESM initial-condition ensembles. By training on multiple ESMs, MESMER is furthermore able to create a multi-model initial condition ensemble, i.e. "super-ensemble" which accounts for uncertainties arising from model formulation. The probability distributions of grid-point level, yearly temperatures generated by MESMER are especially

relevant when used as input data for simulation of impacts that depend on this variable. MESMER thus offers the perspective to improve our understanding of the likelihood of future impacts under multiple scenarios.

Considering the importance of monthly and seasonal information in assessing the impacts of climate change (Zhao et al., 2017; Schlenker and Roberts, 2009; Wramneby et al., 2010; Stéfanon et al., 2012; Pfleiderer et al., 2019), extending the MESMER approach to grid point level monthly temperatures appears desirable. Such holds additional value in assessing the

evolving likelihoods of future impacts, as the temperature response at monthly timescales displays heterogeneities distributive onto seasonal to monthly variabilities and therefore uncertainties, which are otherwise unapparent at annual timescales. In particular, winter months can warm disproportionately more than summer months (Holmes et al., 2016; Loikith and Neelin, 2019; Fischer et al., 2011) which in turn leads to non-stationarity in the amplitude of the seasonal cycle (i.e. intra-annual temperature variabilities) with evolving yearly temperatures (i.e. the intra-annual temperature response is heteroscedastic with

regard to yearly temperature) (Fischer et al., 2012; Huntingford et al., 2013; Thompson et al., 2015; Osborn et al., 2016). Additionally, given that monthly temperature distributions have been observed to display non-Gaussianity, evolving yearly temperatures may cause disproportionate effects on their tail extremes, leading to changes in skewness (Wang et al., 2017; Sheridan and Lee, 2018; Tamarin-Brodsky et al., 2020).

This study focusses on extending MESMER's framework to consist of a local monthly downscaling module (MESMER-M).

This enables the estimation of uncertainty due to natural variability as propagated from annual to monthly timescales since MESMER-M builds upon MESMER, which has already been validated as yielding spatio-temporally accurate variabilities (Beusch et al., 2020). In constructing MESMER-M, we furthermore place emphasis on representing heteroscedasticity of the intra-annual temperature response as well as changes in skewness of individual monthly distributions in a spatio-temporally accurate manner. The structure of this study is as follows: we first introduce the framework of the emulator under Section

3.1 and the approach to verification of the emulator performance under Section 3.2, we then provide the calibration results of the emulator and its example outputs under section 4 and verification results under section 5 after which we proceed to the conclusion and outlook under section 6.



## 2  Data and Terminology

In the analysis, 38 CMIP6 models (Eyring et al., 2016) are considered, using simulations for the SSP5-8.5 scenario high-
emission scenario (O'Neill et al., 2016). Where an ESM's initial-condition ensemble set contains more than one member, it is
split into a training set (used for emulator calibration) and a test set (used for emulator cross-validation). The train-test split
is done in a 70-30 manner. A summary of the CMIP6 models used, their associated modeling groups and the initial condition
ensemble members present within the training and test sets are given in Table A1 in Appendix A. All ESM runs are obtained at
a monthly resolution, bilinearly interpolated to a spatial resolution of $2.5° \times 2.5°$ . The emulator is trained on yearly averaged
temperature values. The term 'temperature' here refers to anomalies of surface air temperature ('tas') relative to the mean
temperature value over the reference period of 1870-1899.

## 3  Methods

### 3.1  MESMER

MESMER is an ESM-specific emulator built to produce spatially resolved, yearly temperature fields by considering both the
local mean response and the local variability surrounding the mean response. Within MESMER, local temperature anomalies
T for a given grid point s and year y are emulated as follows (Beusch et al., 2020):

$$T_{s,y} = g_s(T_y^{glob}) + \eta_{s,y} = \beta_s^{smooth} \cdot T_y^{glob,smooth} + \beta_s^{var} \cdot T_y^{glob,var} + \beta_s^{intercept} + \eta_{s,y} \qquad (1)$$

Where $g_s$ is the local mean response to global mean temperatures $T^{glob}$ and consists of a multivariate, linear regression on
the smooth $T^{glob,smooth}$ (capturing the trend in $T^{glob}$) and variability $T^{glob,var}$ components of $T^{glob}$ with coefficients $\beta_s^{smooth}$
and $\beta_s^{var}$ respectively and intercept term $\beta_s^{intercept}$. $\eta_{s,y}$ represents the local variability surrounding the mean response.

### 3.2  MESMER-M

We divide MESMER-M into a mean response module and a residual variability module, each calibrated on ESM simulation
output data for each grid point individually according to the procedure described in this section and summarised in Figure
1. Such division of a modelling exercise has previously been done in other climate model emulations (Link et al., 2019;
Alexeeff et al., 2018; Tebaldi and Arblaster, 2014; Beusch et al., 2020) and comes with its underlying assumptions. The
primary assumption in our case is that the ESM monthly temperatures are distinctly separable into a mean response component
and a residual variability component. Traditionally the mean response module is designed to be dependent on a certain forcing
(in this case yearly temperatures), while the variability module is space-time dependent. Given the expected changes in monthly
skewness with evolving yearly temperature however, we furthermore propose both a space-time and temperature dependent
variability module. A second assumption made is that other external forcings, such as changes in land cover, have generally





little impact on the monthly temperature response and are therefore not explicitly included in our framework. Given the modular approach we take however, this assumption could potentially be remedied in the future with the addition of separate modules which isolate the signals of such external forcings.

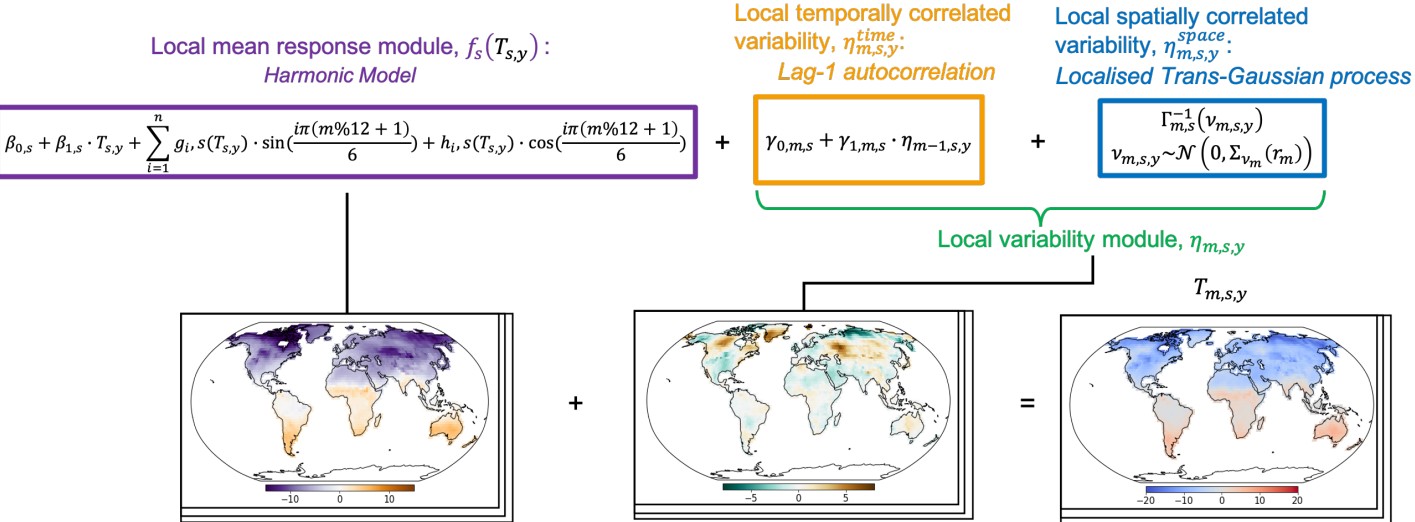

**Figure 1.** Framework for Monthly Emulator

### 3.2.1 Mean response module

The mean response module was conceived to simply but convincingly represent the monthly cycle's mean response to, as well as the changes in the amplitude of the seasonal cycle with yearly temperatures (intra-annual variability) at a grid-point level. To that end, we employ a harmonic model consisting of a Fourier Series, with amplitude terms fitted as linear combinations of yearly temperature, centred around a linear function of yearly temperature as shown in equation 2,

$$f_s(T_{s,y}) = \beta_{0,s} + \beta_{1,s} \cdot T_{s,y} + \sum_{i=1}^{n} \left[ g_{i,s}(T_{s,y}) \cdot sin(\frac{i\pi(m\%12+1)}{6}) + h_{i,s}(T_{s,y}) \cdot cos(\frac{i\pi(m\%12+1)}{6}) \right] \qquad (2)$$

where $T$ is temperature, $m, s$ and $y$ refer to month, grid-point and year indexes, $\%$ is modulo and $g_i$ and $h_i$ are linear functions of $T_{s,y}$ for the $i^{th}$ ordered term of the Fourier Series. An added benefit of using a Fourier Series is that both month-to-month correlations and yearly mean temperatures are well conserved. Since the monthly cycle revolves around its yearly temperature, fitting results for $\beta_0$ and $\beta_1$ coefficients had negligible effects ($\beta_0$=0 and $\beta_1$=1) and for simplicity's sake, we show the Fourier Series as centred around yearly temperature values within the results section. In choosing the order ($n$) of the



harmonic model we sought to have the optimal compromise between model complexity and accuracy. For this, at each grid point we calculated the Bayesian Information Criterion (BIC) of harmonic models fitted with orders $n = 1, \ldots, 8$ and chose the order with the lowest BIC score.

### 3.2.2 Residual variability module

The local residual variability, i.e. the difference between actual local monthly temperature and its mean monthly response to variations in local annual temperature (given by the harmonic model described under section 4.1.1), is assumed to be mostly a manifestation of intra-annual variability processes. It thus can be thought of as short-term spatio-temporally correlated patterns. We follow an approach similar to previous MESMER developments and divide this module further into two parts as shown in equation 3 (Beusch et al., 2020).

$$\eta_{m,s,y} = \eta_{m,s,y}^{time} + \eta_{m,s,y}^{space} \tag{3}$$

Where $\eta_{m,s,y}$ is the total residual variability (otherwise simply called 'residuals') at month, grid-point and year $m, s$ and $y$ and $\eta_{m,s,y}^{time}$ and $\eta_{m,s,y}^{space}$ are its respective time-dependent and space-dependent components.

After a time lag of 1 month, the residual variability is expected to have rapidly decaying covariation such that long term patterns (if any) covary with yearly temperature. $\eta_{m,s,y}^{time}$ is thus modelled using lag-1 autocorrelations as shown in equation 4.

$$\eta_{m,s,y}^{time} = \gamma_{0,m,s} + \gamma_{1,m,s} \cdot \eta_{m-1,s,y} \tag{4}$$

Where $\gamma_{0,m,s}$ and $\gamma_{1,m,s}$ are coefficients fitted per month.

Following this, the rest of the residual variability, represented as $\eta_{m,s,y}^{space}$, requires to not only render spatial correlation structures but also the month-dependent skewness within residuals, expected to be non-stationary with regard to yearly temperatures (see Shapiro-Wilkes test for January and July in Appendix C). Since operating in a Gaussian data space is a prerequisite to infer correlation structures in stochastic modelling, we use a trans-Gaussian process relying on a power transformation to normalise data before sampling from it. This strategy has already been pursued in problems concerning skewness in residual variability (Frei and Isotta, 2019). Precisely, we use the monotonic Yeo-Johnson transformation ($\Gamma_{m,s}$), which accounts for both positive and negative data values, to locally normalise monthly residuals (Yeo and Johnson, 2000). $\Gamma_{m,s}$ relies on a $\lambda$ parameter to deduce the shape of a distribution and normalise accordingly. Non-stationarity in monthly residual skewness with regard to yearly temperature is taken into account by defining the $\lambda$ parameter as a logistic function of yearly temperature, shown in equation 5.

$$\lambda_{m,s,y} = \frac{2}{1 + \xi_{0,m,s} \cdot e^{\xi_{1,m,s} \cdot T_{s,y}}} \tag{5}$$

Where $\xi_{0,m,s}$ and $\xi_{1,m,s}$ are coefficients fitted using maximum likelihood. The performance improvement of having a $\lambda_{m,s,y}$ parameter that is dependent on yearly temperature compared to a case where it is invariant ($\lambda_{m,s}$) is demonstrated by additional tests shown in Appendix D (only shown for January and July).



Earth System
Dynamics



Discussions

$\eta_{m,s,y}^{space}$, is modelled using the transformed residuals through a localised monthly multivariate trans-gaussian process, and thus preserves spatial correlation structures up to a certain distance, as shown in equation 6.

$$\eta_{m,s,y}^{space} \sim \Gamma_{m,s}^{-1}(\mathcal{N}(0, \Sigma_{\nu_m}(r_m))) \text{ with } 1500\text{km} \leq r_m \leq 8000\text{km} \tag{6}$$

Where $\Gamma_{m,s}^{-1}$ is a local inverse Yeo-Johnson transformation and $\mathcal{N}(0, \Sigma_{\nu_m}(r_m))$ is a multivariate trans-gaussian process with means 0 and covariance matrix, $\Sigma_{\nu_m}$, localised using a Gaspari-Cohn function (Gaspari and Cohn, 1999) with radius $r_m$. $r_m$

is month-specific and selected using in a similar cross-validation with a leave-one-out approach as previous MESMER fittings (Beusch et al., 2020) using distances of 1500 km to 8000 km at 250 km intervals. The covariance matrix, $\Sigma_{\nu_m}$, is derived based off the mathematical expectations of the residual variabilities' standard deviations once lag-1 autocorrelations are removed, so as to decouple the spatial component from the temporal component (Matalas, 1967; Richardson, 1981), as shown in Equation 7.

$$\Sigma_{\nu_m}(r_m) = \sqrt{1 - \gamma_{1,m,i}^2} \cdot \sqrt{1 - \gamma_{1,m,j}^2} \cdot \Sigma_{\widetilde{\eta}_m}(r_m) \tag{7}$$

Where $\Sigma_{\widetilde{\eta}_m}$ is the covariance matrix constructed across all grid points for a given month from the locally normalised empirical residuals.

## 3.3    Evaluating emulator performance

### 3.3.1    Mean response verification

To evaluate how well the monthly cycle's mean response, $f_s(T_{s,y})$, is captured we calculate the Pearson Correlation coefficients between the predicted mean response values and their training run values across the whole globe for each month. This not only gives an idea of how well the magnitude of mean response changes correspond to the monthly cycles of the training runs, but also how in phase they are. Where test runs are available, their correlations to the harmonic model results are also calculated to assess how well the harmonic model can represent data it hasn't been trained on. Ideally, the test run correlations should be more

or less equal to those of the training runs, with anything much lower indicating overfitting and anything much higher indicating an under representative training set (i.e. further modifications in the train to test splitting would have to be considered).

### 3.3.2    Residual variability verification

In order to evaluate how well the emulator reproduces the deviations from the harmonic model simulated by the ESMs, 50 emulations are generated per training run. First, we check that short-term temporal features are sufficiently captured: each

residual variability sequence is decomposed into its continuous power spectra, from which we verify, by computing Pearson Correlation coefficients, that the frequency bands with the top 50 highest power spectra within the training run residual variabilities appear with similar power spectra in the corresponding emulated residual variabilities. Second, we verify that the spatial covariance structure is maintained by calculating monthly spatial cross correlations across the residuals produced by each individual emulation and, using Pearson Correlations, comparing them to those of their respective training runs. Where





test runs are available, a similar verification between them and training runs is done, thus yielding an approximation of how actual ESM initial-condition ensemble members should relate to each other.

### 3.3.3 Regional-scale ensemble reliability verification

The emulated monthly temperature values are evaluated across all 26 SREX regions (Seneviratne et al., 2012) for each individual month (see Appendix B for details on SREX regions). We assess how well the emulator can reproduce the 5[th], 50[th]

and 95[th] quantiles of the respective ESM initial-condition ensemble over the periods of 1870-2000 and 2000-2100, by means of quantile deviations following the same approach as Beusch et al. (2020). To do so, we first extract the emulated quantile values for each time step from the distribution spanning the full set of emulations at that time step, thus creating smooth time series of the emulated quantiles. For each of the above named time periods, we then compute the ESM's quantile respective to the emulated quantile as the proportion of time steps that the ESM training runs- and where available test runs -are below

the emulated quantile values. The deviation of the ESM computed quantile from the actual emulated quantile is then assessed, where positive deviations mean that the emulated quantile is warmer than that of the ESM run and vice versa.

## 3.4 Benchmarking MESMER-M using a simple physical approach

MESMER-M is designed to provide a purely statistical representation of monthly temperatures in a spatially resolved manner, taking only local yearly temperatures as input. Intra-annual processes which are otherwise unapparent on yearly timescales

may however emerge on monthly timescales, introducing additional variability and -due to possible secondary biophysical feedbacks- month and season specific non-uniformities (e.g. non-linearities, non stationarities, changes in distributional properties) (Potopová et al., 2016; Xu and Dirmeyer, 2011; Jaeger and Seneviratne, 2011; King, 2019; Tamarin-Brodsky et al., 2020) into the monthly temperature response. As described in Section 3.1.2 and following existing downscaling theory (Berner et al., 2017; Arnold, 2001), such variability is stochastically accounted for within MESMER-M's local residual variability module

by sampling month-specific, spatially correlated variability terms from a transformed Gaussian space, where non-uniformities are somewhat also inferred by allowing the skewness to covary with local yearly temperatures. In this section, we delineate a framework to verify that this statistical approach, based on a single input variable of yearly temperature, can sufficiently imitate properties the monthly temperature response which otherwise result from secondary biophysical feedbacks.

To isolate the contribution of secondary biophysical feedbacks to the monthly temperature response, we consider them as

inducing the residual differences between the ESM and harmonic model realisations. This follows from the harmonic model representing the expected direct mean response to evolving yearly temperatures, with any systematic departure from it being driven by secondary forcers. To rudimentarily represent these contributions, a simple, physical model consisting of a suite of Gradient Boosting Regressors (GBRs) (Hastie et al., 2009) is built for each ESM. Each GBR within the suite represents one grid point and is trained to predict the local residual differences using local biophysical variable values (see Table 1) as predictors.

The list of predictors is complemented by local yearly temperature values and month values in their harmonic form (hence



$\frac{\pi(1\%12+1)}{6}$ for January, etc.) to account for month dependencies in residual variabilities and yearly temperature influences (if any) left behind within the residuals.

To optimise the selection of the biophysical variables used as predictors, we first compare the performance of different physical models trained using different sets of biophysical variables for each ESM. Pearson Correlations calculated over all months, between ESM test runs and harmonic model test results augmented by biophysical variable based physical model predictions relative to those obtained when augmenting using only $T_{yr}$ and month based physical model predictions are used as a measure of performance. This additionally allows determination of whether improvement in residual representation comes from the added biophysical variable information and if so where and how. The best globally performing model is selected as a benchmark to assess how well the residual variability module, described in Section 3.1.2, statistically represents properties within the monthly response arising from secondary biophysical feedbacks. As we are most interested in the representation of monthly temperature distributions and the influences of biophysical feedbacks therein, we compare the energy distances from the actual ESM runs of the benchmark, "physical" emulations -constituting the mean response with GBR predicted residuals added ontop- to those of the statistical emulations -constituting the mean response with residuals from the residual variability module added ontop (as described in Section 3.1). The energy distance is a non-parametric estimate of the distance between two cumulative distribution functions (cdfs), x and y, by considering all their independent pairs of variables, $\{X_i, X_j\}$ and $\{Y_k, Y_l\}$ respectively:

$$D(x,y) = (2E||X_i - Y_k|| - E||X_i - X_j|| - E||Y_k - Y_l||)^{\frac{1}{2}} \tag{8}$$

Time series of the biophysical variables are obtained from CMIP6 runs. For this analysis, we only focus on ESMs which provided data for all 5 biophysical variables under consideration, for both the test and training runs used during emulator calibration.

**Table 1.** List of Biophysical Variables used in training the Gradient Boosting Regressor

| | Variable | Abbreviation |
|---|---|---|
| | Albedo | A |
| | Snow Cover Fraction (%) | S |
| Biophysical | Cloud Cover Fraction (%) | C |
| | Sensible Heat Flux (Wm$^{-2}$) | Hs |
| | Latent Heat Flux (Wm$^{-2}$) | Hl |
| Other | Yearly Temperature | $T_{yr}$ |
| | Month ($\frac{\pi(1\%12+1)}{6} ... \frac{\pi(12\%12+1)}{6}$) | month |



# 4 Illustration of emulator attributes

## 4.1 Calibration results

When calibrating the harmonic model constituting the mean response module, highest orders of the Fourier series were found in tropical to sub-tropical regions where the seasonal cycle contains a relatively small shift in temperature values (first row, Figure 2). The Arctic also displays relatively high orders chosen within the Fourier series, possibly due to higher variabilities in the response of the seasonal cycle shape with increasing yearly temperatures. In contrast, temperate regions which possess distinctly sinusoidal seasonal cycles with marked snow-driven summer to winter transitions display relatively lower orders. CanESM5 and MIROC6 show the overall highest orders, this can be traced back to the availability of significantly more training runs, hence more information on which to train the emulator, for these two ESMs (refer to Table A1).

The residual variability module calibration results are shown in Figure 2 for January and July (other months are available upon request). The lag-1 auto-correlation coefficients ($\gamma_{1,m,s}$) mostly exhibit positive values across all ESMs for January, with at least 70% of grid points having values between 0 and 0.3, suggesting minimal month-to-month memory ontop of the seasonal cycle. Apart from some ESMs which are more normally distributed in $\gamma_{1,m,s}$, July shows similar behaviour albeit with a larger spread in values. HadGeM3-GC31-LL and ACCESS-CM2 present themselves as outliers here with the bulk of their $\gamma_{1,m,s}$ coefficients centred around 0 for both January and July, indicating negligible autocorrelations.

The average Yeo-johnson lambda parameter ($\widetilde{\lambda}_{m,s}$) displays a shift of values greater than 1 to values close to 1 in the Northern Hemisphere (30°-50°) between January and July. In general, $\widetilde{\lambda}_{m,s}$ values greater (less) than 1 indicate a concave (convex) transformation function owing to negative (positive) skewness, while values equal to 1 suggest minimal skewness in the input distribution (Yeo and Johnson, 2000). This explains the seasonality in $\widetilde{\lambda}_{m,s}$ as we expect a more negatively skewed residual distribution in the winter when the snow-albedo feedback leads to a non-linear winter-time warming (Cohen and Rind, 1991; Hall, 2004; Colman, 2013; Thackeray et al., 2019) causing the harmonic model to overestimate the mean temperature response. July displays significantly high $\widetilde{\lambda}_{m,s}$ values for polar latitudes (>80°) explainable by the sudden increase in warming rates during ice-free summers (Blackport and Kushner, 2016). Around the equator (-5° to 5°) we see $\widetilde{\lambda}_{m,s}$ values close to but slightly higher than 1, with INM-CM5-8 and INM-CM5-0 displaying significantly high values. The source of this varies model to model but mainly originates from the North-West South America and Sahel regions, alluding to the presence of some non-linear warming response in these regions.

Localisation radii vary from model to model and are generally higher in January than July reflecting seasonal differences in residual behaviour possibly due to winter snow cover yielding larger spatial patterns. CanESM5 and MIROC6 display notably higher localisation radii, which can again be tracked back to them having more training runs: more information is available in the covariance matrix constructed, thus higher localisation radii perform well in fitting using the leave-one-out approach.

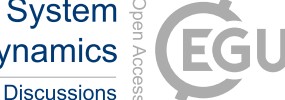

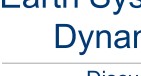

**Figure 2.** Calibration parameters obtained from emulator fittings for all CMIP6 models. For the mean response module, the highest order of the Fourier Series considered in the harmonic model is plotted against latitude (row 1). For the monthly residual variability module, parameters are displayed for January (rows 2-4) and July (rows 5-7). The local lag-1 autocorrelation coefficients are plotted as boxplots (rows 2 and 5) with whiskers covering the 0 to 100 quantile range, $\lambda_{m,s,y}$ coefficients averaged over all years for the local yeo-johnson transformation are plotted against latitude (rows 3 and 6) and the localisation radii are given as bar charts (rows 4 and 7).

## 4.2 Regional behaviour for a select 4 ESMs

Figure 3 visually demonstrates the harmonic model (bold black lines) capturing the mean monthly temperature response for both January and July, at global and regional scales (here we show the SREX regions WNA and WAF), across all 4 ESMs. The





remaining natural variability surrounding the mean response displays a month dependency across all ESMs, such that January
variabilities are up to double that of July both globally and in the displayed regions. These month dependencies in variabilities
are well accounted for within the full emulations (grey lines) comprising both the harmonic model and the residual variability
module, justifying the prescription of a month specific residual variability module.

Figure 4 shows the trends in the intra-annual temperature variability with evolving yearly temperatures. The harmonic model
is able to capture the general trends displayed by the ESMs, albeit not being able to fully account for non-linearities within
them (for example for MPI-ESM1-2-LR in WAF, where an increase in variability followed by stabilisation is observed, and
is represented by the harmonic model as a linear increase with no stabilisation). This suggests the presence of secondary,
non-linear processes which cannot be fully accounted for by yearly temperature only. Where the harmonic model falls short
in representing non-linearities in the response of the seasonal cycle's amplitude, the addition of residual variability to generate
full emulations is nevertheless able to capture the overall spread.

## 5 Evaluating emulator performance

### 5.1 Mean response verification

Pearson Correlations between the harmonic model and ESM training runs range from 0.7 up to almost 1 (Figure 5). Summer
months exhibit the highest correlations while transition months of spring and autumn have the lowest correlations. Such low
correlations could result from the ESM initial-condition ensemble spread in representing the timing of snow cover decrease
and increase, such that the mean response extracted does not fully match any individual ensemble member alone. Winter month
correlations are generally higher than those of transition months but lower than those of summer months. This is possibly due to
snow-albedo feedbacks, which induce non-linearities into the winter period mean response (Cohen and Rind, 1991; Hall, 2004;
Colman, 2013; Thackeray et al., 2019). Hence, even though the extracted mean response corresponds well to the individual
ensemble members, correlations are still lower than those of summer months where the response is more linear. Overall the
training run correlations correspond well to test run correlations (where available) confirming good data representation within
the training set and minimal (if any) model overfitting.





**Figure 3.** Regionally averaged temperature time series (rows) of January and July for four example ESMs (columns). Temperature values are given as anomalies with respect to the mean temperature value over the reference period of 1870-1899. The regions are from top to bottom: global land without Antarctica, West North America (WNA), and West Africa (WAF). Harmonic model values are plotted in black, full emulator values in grey and ESM runs in colour.



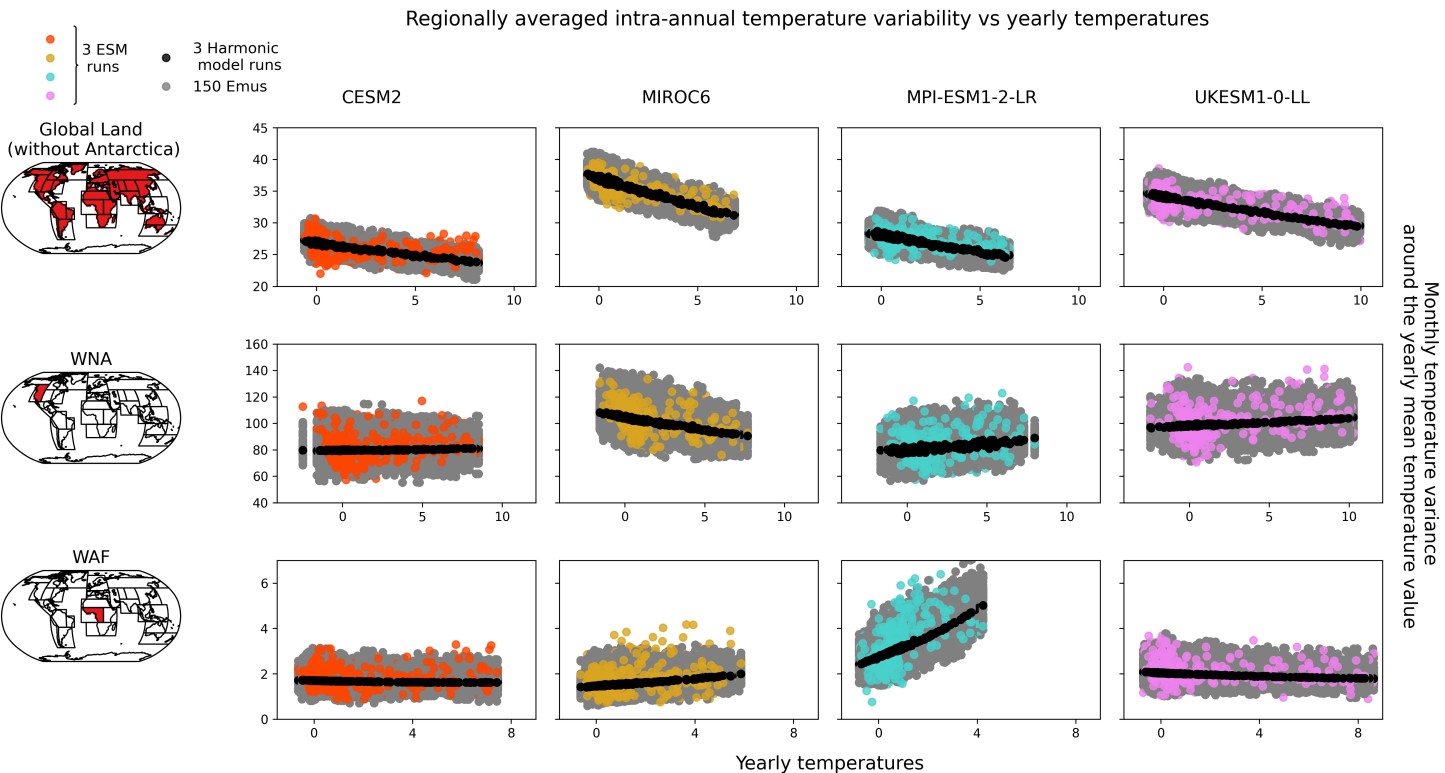

**Figure 4.** Regionally averaged intra-annual temperature variability (i.e. variation of each year's monthly temperatures around the yearly mean) scatter plotted against yearly temperature (rows) for four example ESMs (columns). Temperature values are taken as anomalies with respect to the mean temperature value over the reference period of 1870-1899. Each dot represents the temperature variance calculated from the monthly values for one individual year. The regions are from top to bottom: global land without Antarctica, West North America (WNA) and West Africa (WAF). Harmonic model values are plotted in black, full emulator values in grey and ESM runs in colour.



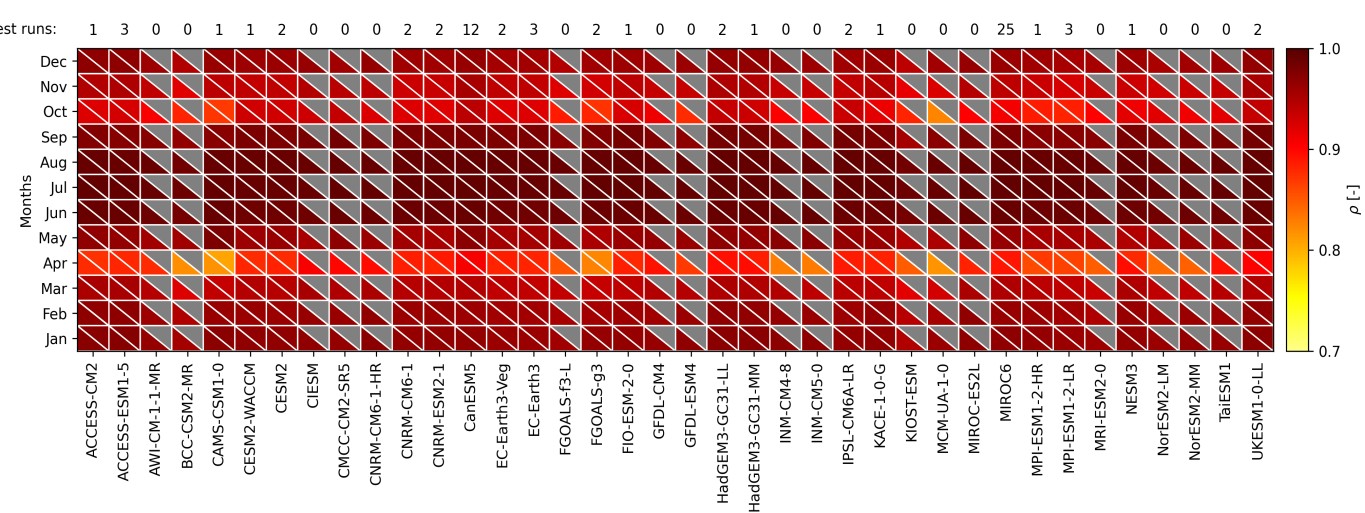

**Figure 5.** Local mean monthly response verification for the select CMIP5 models and all CMIP6 models by means of Pearson Correlation between the harmonic model and training runs (indicated by the colour of the lower triangle), over all global land grid points (without Antarctica) for each month. The correlations between the harmonic model and test runs are given in colour in the upper triangles to see how well the harmonic model performs for data it has not seen yet (a grey upper triangle means that no test run is available for this model). The number of test runs for each model is given on the upper x-axis labels .



## 5.2 Residual variability verification

To establish that temporal patterns within the ESM residual variabilities are successfully emulated, their respective power spectra are considered. Results shown in Figure 6 display the emulator's median correlations lying between 0.8 and 0.93. This

corresponds well - although is slightly lower than - the correlations across the ESM test runs (crosses) which are between 0.88 and 0.95. Correlations across emulations for a given ESM display very little spread, which is in agreement with the near identical correlations seen amongst ESM test runs. In the example 2D histogram plot (given for CESM2), we see that the emulator trained on that ESM is able to capture lower power spectra, however, underestimates higher power spectra. This is a consequence of the emulator design, as we restrict ourselves to considering only lag-1 autocorrelations such that higher order

temporal patterns are not accounted for. Nevertheless, the occurrence of such higher power spectra is low for the considered ESM and the overall bulk of the power spectra correlate well,

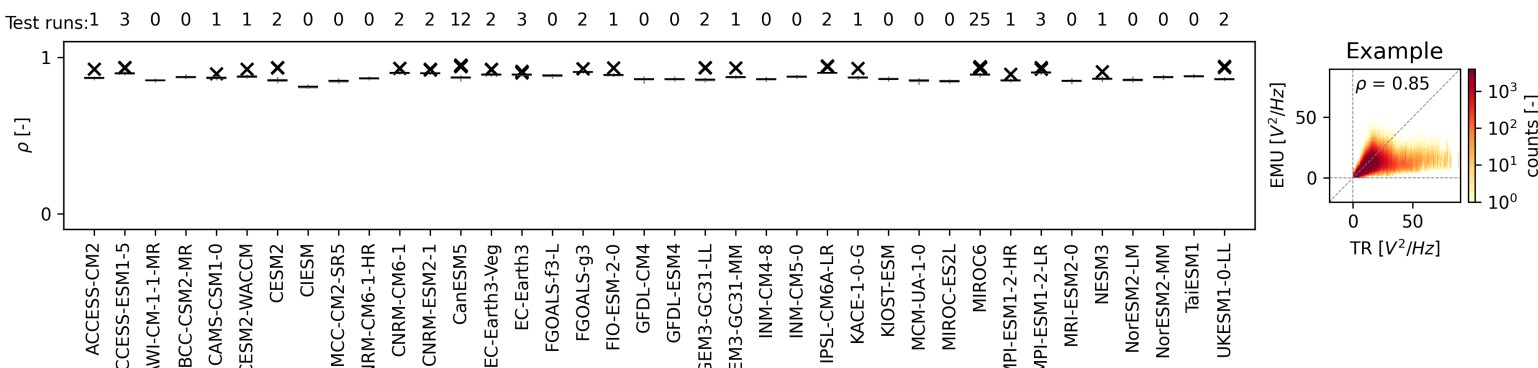

**Figure 6.** Verification of the time component of the local variability module by means of Pearson Correlations between the power spectra of the top 50 frequency bands present within the full training runs (i.e. considering all months together) and the power spectra at which they appear within the full emulations (box plots). Where test runs are available, as indicated by the numbers in the upper x-axis, their correlations with training runs are also given (black crosses). The example 2D histogram shows the power spectra for CESM2 training runs versus the corresponding power spectra within its emulations.

For verification of the residual variability's spatial component, we consider the spatial cross-correlations within 4 geographical bands centred around the grid cell for which temperature is being emulated (Figure 7) for each month (here we show only January and July). As the spatial covariance matrix within the emulator is localised (see Section 3.1.2), its spatial cross-

correlations are by design expected to diminish with increasing distances. Hence, we see the emulator performing best at distances below 1500 km, with median correlations of 0.91-0.99 which are in line with those of the ESM test runs (crosses). Beyond 1500 km, the emulator performs progressively worse with correlations dropping below 0.1 for distances between 3000 km and 6000 km and staying there for distances larger than 6000 km, while those of test runs remain around 0.5-0.8. CanESM5




and MIROC6 are the two exceptions at distances between 3000 km and 6000 km, with correlations of 0.33-0.5 which then again
drop to below 0.1 at distances larger than 6000 km. This is due to their notably larger localisation radii (see Figure 2), which
leads to a slower decline of spatial cross-correlations with increasing distances as compared to other ESMs.

## 5.3    Regional-scale ensemble reliability verification

Regionally aggregated 5th, 50th and 95th quantile deviations of the ESM training (and where available test) runs from the full
emulations (consisting of both the mean response and local variability module) are plotted for January (Figures 8 and 10) and
July (Figures 9 and 11) for the periods of 1870-2000 and 2000-2100. The 50th quantile deviations over the period of 1870-2000
in January and July (Figures 8 and 9, respectively) generally show low values (-3% to 3%). A slight regional dependency for
this period is visible, where tropical/sub-tropical regions of AMZ, NEB, SSA, WAF, EAF, SAF have generally warmer (colder)
emulated 50th quantiles as compared to the ESM runs, while those of the remaining regions are colder (warmer) for January
(July). While January 50th quantile deviations over the period of 2000-2100 remain low with less (if any) distinguishable
regional dependency, July 50th quantile deviations for this period increase (-10% to 10%) with an opposite pattern in regional
dependency to that of 1870-2000. The increase for July in deviations could be a combined result of non-linear warming and
relatively lower variability in July temperature values as compared to those of January in the ESM simulations. Hence, the
linear emulated mean response diverges from the ESM median warming, and the addition of residual variability cannot fully
resolve this due to the overall higher signal to noise ratio. This shows a limitation in the emulator's design, where delegating
the representation of secondary, non-linear responses to the residual variability module does not fully work in the presence of
lower variabilities.

Generally, emulated 5th (95th) quantiles are warmer (colder) than those of the ESM training and test runs. Such under-
dispersivity for regional averages is linked to the localisation of the spatial covariance matrix within the residual variability
module, such that spatial correlations drop faster within the emulator than they do in the actual ESM. Underdispersivity may
also be a consequence of the emulator solely relying on yearly temperatures as an input variable. To be exact, even though
changes in skewness in local monthly temperature residuals with evolving local yearly temperatures is incorporated, such can-
not completely represent delayed or compound effects, otherwise brought about by secondary/external feedbacks (Shinoda,
2001; Potopová et al., 2016), which more largely affect extreme quantiles (Huybers et al., 2014; Guirguis et al., 2018; Loikith
and Neelin, 2019). January 5th and 95th quantile deviations over the time period of 1870-2000 show lowest values in the South-
ern Hemispheric regions of AMZ, NEB, WSA, SSA and SAF, with emulations sometimes even being slightly overdispersive.
For July of the same period, this behaviour switches to most Northern Hemispheric regions but only for select ESMs. For the
same select ESMs, similar features as seen in January and July for 1870-2000 are seen again for both months in the period of
2000-2100. An inspection of the localisation radii used within these select ESMs points at generally high values, explaining
the higher dispersivity as spatial cross correlations are conserved up to larger distances.






Figure 7. Verification of the spatial representation within the local variability module. Pearson correlations between ESM training run and emulated spatial cross-correlations are considered for 4 geographical bands centred around the grid cell for which temperature is being emulated (rows) at individual months of January (black boxplots) and July (red boxplots). Where ESM test runs are available, as indicated by the numbers on the upper x-axis, their correlations with training runs are also given for January (black crosses) and July (red crosses). Example 2D histograms of the January spatial cross-correlations for CESM2 training runs versus those of its emulations are given for each geographical band.



## 1870-2000 January regional-scale verification: deviation of climate model runs from emulated quantiles



**Figure 8.** January 5% (left), 50% (middle), and 95% (right) quantile deviations (colour) of the monthly emulated quantile from that of its ESM training (top block) and test (bottom block) runs, over the period 1870-2000 for Global land (without Antarctica) and SREX regions (columns) across all CMIP6 models (rows). The monthly emulated quantile is computed based on 50 emulations per ESM run and quantile deviations are given as averages across the respective number of ESM training/test runs. The number of test runs averaged across is indicated in brackets next to the model names in the bottom block. Red means that the emulated quantile is warmer than the quantile of the ESM run, and vice versa for blue.





**Figure 9.** Same as Figure 8, but for July over the period 1870-2000

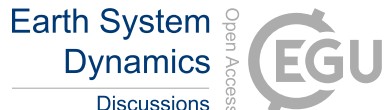

**Figure 10.** Same as Figure 8, but over the period 2000-2100



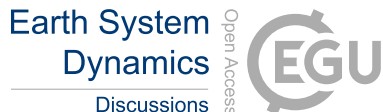

**Figure 11.** Same as Figure 8, but for July over the period 2000-2100





## 5.4 Benchmarking MESMER-M using a simple physical approach

In-depth analysis of the benchmarking approach, based on a suite of Gradient Boosting Regressors using biophysical variables as predictors, outlined in Section 3.3 is conducted for 4 select ESMs which exhibit diverse genealogies (Knutti et al., 2013; Brunner et al., 2020) (see Figure E1 in Appendix E for summarised results of all other ESMs). From Figure 12, it is evident that adding even one biophysical variable explains part of the residual difference behaviour, with correlations over global land always being positive. Across all 4 ESMs the main improvements are in Northern Hemispheric regions which possess distinct seasonal variations in snow cover namely, ALA, CGI, WNA, CNA, ENA, NEU, CEU, NAS and CAS. MIROC6 and MPI-ESM1-2-LR exhibit additional improvements in other regions, notably WAF, EAF, SAS and NAU regions. It is worth noting that for most ESMs, the biophysical predictor configuration of albedo, cloud cover and snow cover (ACS) performs consistently worse than any configuration containing sensible and latent heat fluxes (H). This suggests the presence of colinearities between albedo, snow cover and cloud cover, which limits the physical model's representation of certain residual variability behaviour, otherwise explainable using sensible and latent heat fluxes. MIROC6 is an exception to this, with both the biophysical configurations of latent heat flux (Hl) and H yielding 0 or lower relative correlations in WNA, CNA, ENA, CEU and EAS while HC displays no improvements for these regions. This could be due to colinearities between cloud cover and latent heat fluxes alongside overfitting of the physical model to latent heat fluxes due to confounding variabilities.

As HACS performs the best globally (i.e. appears as 1 in global land) across all 4 ESMs we choose it as the benchmark physical model to compare the residual variability module to. Figure 13 shows the energy distances of the physical (harmonic model+HACS) and statistical (full emulator) emulated cdfs to the ESM cdfs for January and July (other months available on request), where 0 indicates identical, and thus 'perfect' emulated cdfs. Energy distances in July for both the physical and statistical models are close to 0 indicating near perfect cdfs, with only MIROC6 and MPI-ESM1-LR showing larger distances for the full emulator in the Indo-Gangetic region, South America and Central-West Africa. In contrast, January shows higher distances for both the physical and statistical model cdfs, particularly in Northern Hemispheric regions with seasonal snowfall and most notably in the full emulator of MIROC6. Overall, the statistical model performs better than the physical model for CESM2 and UKESM1-0-LL and worse for MIROC6 and MPI-ESM1-2-LR. An explanation behind this could be that MIROC6 and MPI-ESM1-2-LR have at least 4 more training runs than CESM2 and UKESM1-0-LL, providing the GBR model with more training material to extract biophysical information from. This suggests a limit to when the statistical approach performs better than the physical approach, depending on how much information is available to train on. Nevertheless, without the prerequisite of having more training runs -which can be seen as an added advantage- the statistical approach taken by the full emulator generally shows better performances across most ESMs for January and July than the physical approach (Figure E2 in Appendix E). Thus, the distributional properties of local monthly temperatures as seen within ESM initial-condition ensembles can be sufficiently represented using the statistical approach outlined in this paper, which takes only local yearly temperatures as input.



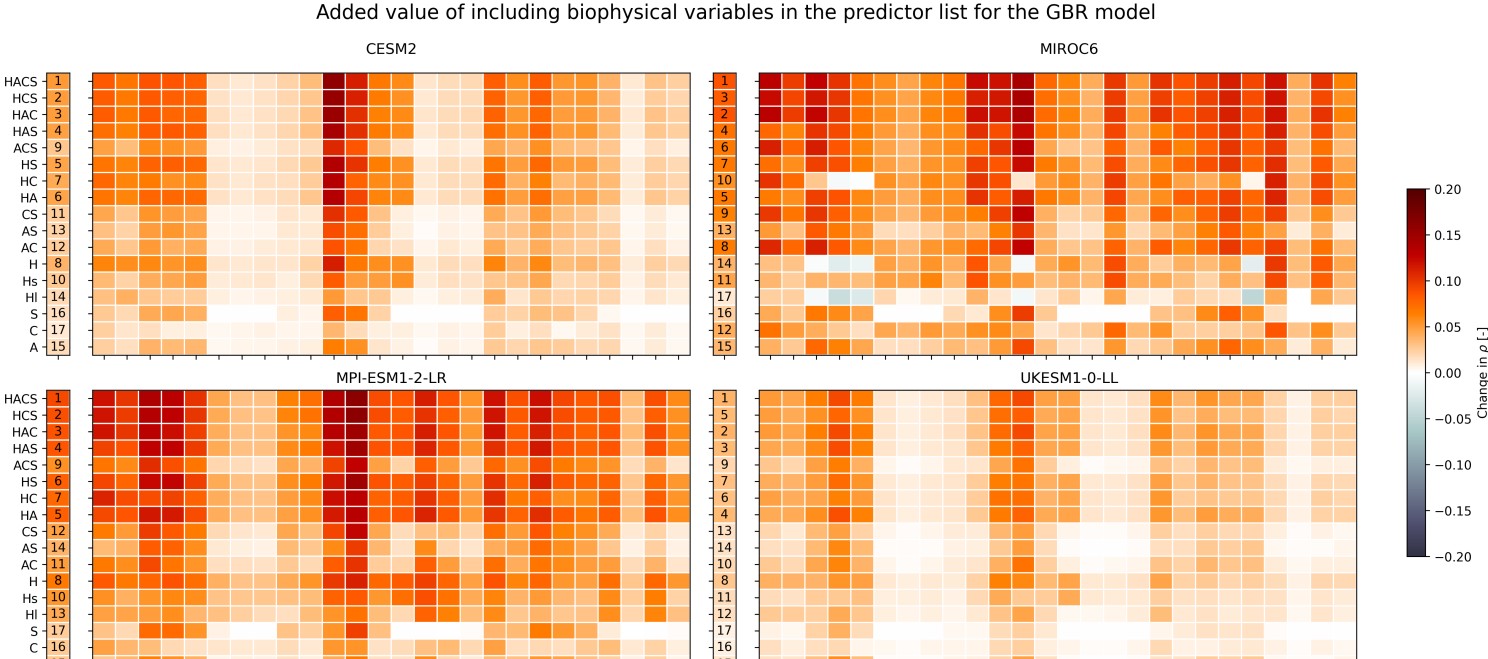

**Figure 12.** Global land (without Antarctica) and regional performances (columns) of the physical model trained using different predictor sets (rows) shown for 4 select CMIP6 models, for each SREX region. Acronyms for the predictor sets (y-axis tick labels) can be referred back to in Table 1. Pearson Correlations calculated over all months between test runs and harmonic model test results augmented by the physical model's predictions of residual variability are considered. Here we show changes in correlations relative to those obtained when augmenting using only $T_{yr}$ and month values as predictors. Numbers in the global land column indicate the ranking of each predictor set, where 1 is the best performing.



**Figure 13.** Comparison of the performance of the harmonic model + physical HACS model to that of the full emulator for January and July of 4 select CMIP6 models. The energy distance from the actual model test runs is considered, where 0 indicates the best performance.



# 6 Conclusion and Outlook

We extend MESMER's framework to include a monthly downscaling module trained for each ESM at each grid-point individually, thus providing realistic, spatially explicit monthly temperature fields from yearly temperature fields in a matter

of seconds. We assume a linear response of the seasonal temperature cycle to its yearly mean values and represent it using a harmonic model. Any remaining response patterns are expected to arise from secondary biophysical feedbacks which act on regional levels and intra-annual timescales and have asymmetric, non-uniform (e.g. non-linear, non-stationary, affecting variance and skewness) effects across months. To capture them, we build a month-specific residual variability module which samples spatio-temporally correlated terms, thus conserving lag-1 autocorrelations and spatial cross-correlations whilst accounting for

specificities in the residual variability structure across months. By letting the skewness of the residual sampling space covary with yearly temperatures, non-uniformities in secondary feedbacks are furthermore inferred through their manifestations within the monthly temperature distributions.

Verification results across all ESMs show the emulator altogether reproducing the mean monthly temperature response, as well as conserving temporal and spatial correlation patterns and regional-scale temperature distributions up to a degree

sensible to its simplicity. To further assess how well the emulator is able to represent heterogeneities in the monthly temperature response arising from secondary biophysical feedbacks, we compare its performance to that of a simple physical model built on biophysical information. The emulator overall reproduces the cdfs of the actual ESM just as well as, and in most cases even better than the physical model, evidencing the validity of such a statistical approach in inferring temperature distributions and insofar the uncertainty due to natural variability within temperature realisations. Given that the uncertainty due to natural

variability is a property intrinsic to climate models and largely irreducible (Deser et al., 2020), the emulator thus proves itself as a pragmatic alternative to otherwise having to generate large single-model, initial-condition ensembles.

This study demonstrates the advantage of constructing a modular emulator such that the emulator framework can be extended according to the area of application. Additional module developments which increase the impact relevance of the emulations and improve the fidelity in global and regional representation under different climate scenarios should be given priority. A

module that comes to mind would be one representing changes in land cover, such as de/afforestation, which have been historically assessed to have biophysical impacts of a similar magnitude on regional climate as the concomitant increase of greenhouse gases (GHGs) (De Noblet-Ducoudré et al., 2012) and for which very distinct imprints on the seasonal cycle of temperatures as well as the tails of the temperature distributions have been identified (Pitman et al., 2012; Lejeune et al., 2017). Such a module would furthermore increase the emulator's relevance towards impact assessments, in light of the important

land-cover changes expected to happen in the 21st century (Popp et al., 2017; O'Neill et al., 2016) and the relevance of accounting for their regional climate impacts especially in high-mitigation scenarios such as those compatible with the 1.5°C long-term temperature goal of the Paris Agreement (Seneviratne et al., 2018; Roe et al., 2019; Arneth et al., 2019). One technical advantage of adding a land cover module would be that the effect of land cover changes can be expected to be sufficiently decoupled from the overall GHG induced temperature response. Hence, the direct local effect of such a module

would not interfere with the mean temperature response as extracted within the rest of the emulator.





Beusch et al. (2020) suggest that the ESM-specific emulator calibration results represent distinct "model IDs", containing scale-dependent information of the model structure. As a follow-up from the physical model based benchmarking done within this study, we further propose that the residuals from the mean response module also contain ESM-specific, scale-dependent information, constituting the distinct representations and parameterisations of biophysical feedbacks within each ESM. For ex-
ample, models with strong snow-albedo feedbacks and a large snow cover reduction with increasing global mean temperature will show stronger warming of cold months (Fischer et al., 2011) and thus more negatively skewed residuals for those months. A step towards disentangling such process representation within the ESMs has already been made in this study, through the representation of biophysical contributions within residual variabilities using the GBR based physical model. Further analysing the strength of the co-variations of different biophysical variables with the residuals, as identified by the physical model, could then
help isolate the exact contributions of these variables. While the key physical variables contributing to temperature variability within ESMs have already been studied (Schwingshackl et al., 2018), such an analysis would further provide information on the amount by which a choice number of biophysical variables contribute to residual variability within each ESM. Performing a similar analysis on observational datasets and comparing the results to those of the ESMs, could then serve as a means to evaluate model representation of biophysical interactions under a changing climate.



# Appendix A

# Appendix B

Where regional results are shown, we consider area weighted averages for the 26 SREX regions (Seneviratne et al., 2012) as shown in Figure B1. Global land results always constitute area weighted averages across all land grid points excluding Antarctica.

## SREX region abbreviations

**Figure B1.** Map of the SREX regions and their abbreviations. The considered land grid points are shown in grey.



**Table A1.** List of the 38 employed CMIP6 models, the modeling groups providing them, and the number of initial-condition ensemble members used in the training and test sets.

| Model | Modeling Center (or Group) | Training Runs | Test Runs |
|---|---|---|---|
| ACCESS-CM2 | Commonwealth Scientific and Industrial Research Organization (CSIRO) and Bureau of Meteorology (BOM), Australia | 2 | 1 |
| ACCESS-ESM1-5 | Commonwealth Scientific and Industrial Research Organization (CSIRO) and Bureau of Meteorology (BOM), Australia | 7 | 3 |
| AWI-CM-1-1MR | Alfred Wegener Institute, Helmholtz Centre for Polar and Marine Research | 1 | 0 |
| BCC-CSM2-MR | Beijing Climate Center, China Meteorological Administration | 1 | 0 |
| CAMS-CSM1-0 | Chinese Academy of Meteorological Science | 1 | 1 |
| CanESM5 | Canadian Centre for Climate Modeling and Analysis | 12 | 12 |
| CESM2-WACCM | National Center for Atmospheric Research | 2 | 1 |
| CESM2 | Community Earth System Model Contributors | 3 | 2 |
| CIESM | Community Earth System Model Contributors | 1 | 0 |
| CMCC-CM2-SR5 | Centro Euro-Mediterraneo per I Cambiamenti Climatici | 1 | 0 |
| CNRM-CM6-1-HR | Centre National de Recherches Météorologiques / Centre Européen de Recherche et Formation Avancée en Calcul Scientifique | 1 | 0 |
| CNRM-CM6-1 | Centre National de Recherches Météorologiques / Centre Européen de Recherche et Formation Avancée en Calcul Scientifique | 4 | 2 |
| CNRM-ESM2-1 | Centre National de Recherches Météorologiques / Centre Européen de Recherche et Formation Avancée en Calcul Scientifique | 3 | 2 |
| EC-Earth3-Veg | EC-EARTH consortium | 2 | 1 |
| EC-Earth3 | EC-EARTH consortium | 7 | 3 |
| FGOALS-f3-L | LASG, Institute of Atmospheric Physics, Chinese Academy of Sciences and CESS,Tsinghua University | 1 | 0 |
| FGOALS-g3 | LASG, Institute of Atmospheric Physics, Chinese Academy of Sciences and CESS,Tsinghua University | 3 | 1 |
| FIO-ESM-2-0 | The First Institute of Oceanography, SOA, China | 2 | 1 |
| GFDL-CM4 | NOAA Geophysical Fluid Dynamics Laboratory | 1 | 0 |
| GFDL-ESM4 | NOAA Geophysical Fluid Dynamics Laboratory | 1 | 0 |
| HadGEM3-GC31-LL | Met Office Hadley Centre | 3 | 1 |
| HadGEM3-GC31-MM | Met Office Hadley Centre | 2 | 1 |
| INM-CM4-8 | Institute for Numerical Mathematics | 1 | 0 |
| INM-CM5-0 | Institute for Numerical Mathematics | 1 | 0 |
| IPSL-CM6A-LR | Institut Pierre-Simon Laplace | 4 | 1 |
| KACE-1-0-G | National Institute of Meteorological Sciences/Korea Meteorological Administration | 2 | 1 |
| KIOST-ESM | Korea Institute of Ocean Science and Technology | 1 | 0 |
| MCM-UA-1-0 | Department of Geosciences, University of Arizona | 1 | 0 |
| MIROC-ES2L | Japan Agency for Marine-Earth Science and Technology, Atmosphere and Ocean Research Institute (The University of Tokyo), and National Institute for Environmental Studies | 1 | 0 |
| MIROC6 | Atmosphere and Ocean Research Institute (The University of Tokyo), National Institute for Environmental Studies, and Japan Agency for Marine-Earth Science and Technology | 25 | 25 |
| MPI-ESM1-2-HR | Max-Planck-Institut für Meteorologie (Max Planck Institute for Meteorology) | 1 | 1 |
| MPI-ESM1-2-LR | Max-Planck-Institut für Meteorologie (Max Planck Institute for Meteorology) | 7 | 3 |
| MRI-ESM2-0 | Meteorological Research Institute | 1 | |
| NESM3 | Nanjing University of Information Science and Technology | 1 | 1 |
| NorESM2-LM | Norwegian Climate Centre | 1 | 0 |
| NorESM2-MM | Norwegian Climate Centre | 1 | 0 |
| TaiESM1 | Research Center for Environmental Changes, Academia Sinica | 1 | 0 |
| UKESM1-0-LL | Met Office Hadley Centre | 3 | 2 |



**Appendix C**

**Figure C1.** Shapiro-Wilkes test for normality of January temperature residuals. The null hypothesis is that the residuals are normally distributed. A multiple test correction is applied to the p-values before plotting them. Percentage values next to the model names indicate the percentage of land grid points for which the null hypothesis is rejected.

**Figure C2.** Shapiro-Wilkes test for normality of July temperature residuals. The null hypothesis is that residuals are normally distribution. A multiple test, Benjamini/Hochberg correction is applied to the p-values before plotting them. Percentage values next to the model names indicate the percentage of land grid points for which the null hypothesis is rejected.





# Appendix D

**Figure D1.** Likelihood ratio test comparing the performance of January's Yeo-Johnson transformations when using just one single lambda parameter ($\lambda_{m,s}$) vs when using a yearly temperature dependent lambda parameter ($\lambda_{y,m,s}$). The null hypothesis is that the $\lambda_{m,s}$ based transformation performs better than the $\lambda_{y,m,s}$ based transformation. A multiple test, Benjamini/Hochberg correction is applied to the p-values before plotting them. Percentage values next to the model names indicate the percentage of land grid points for which the null hypothesis is rejected.





**Figure D2.** Likelihood ratio test comparing the performance of July's Yeo-Johnson transformations when using just one single lambda parameter ($\lambda_{m,s}$) vs when using a yearly temperature dependent lambda parameter ($\lambda_{y,m,s}$). The null hypothesis is that the $\lambda_{m,s}$ based transformation performs better than the $\lambda_{y,m,s}$ based transformation. A multiple test, Benjamini/Hochberg correction is applied to the p-values before plotting them. Percentage values next to the model names indicate the percentage of land grid points for which the null hypothesis is rejected.





# Appendix E

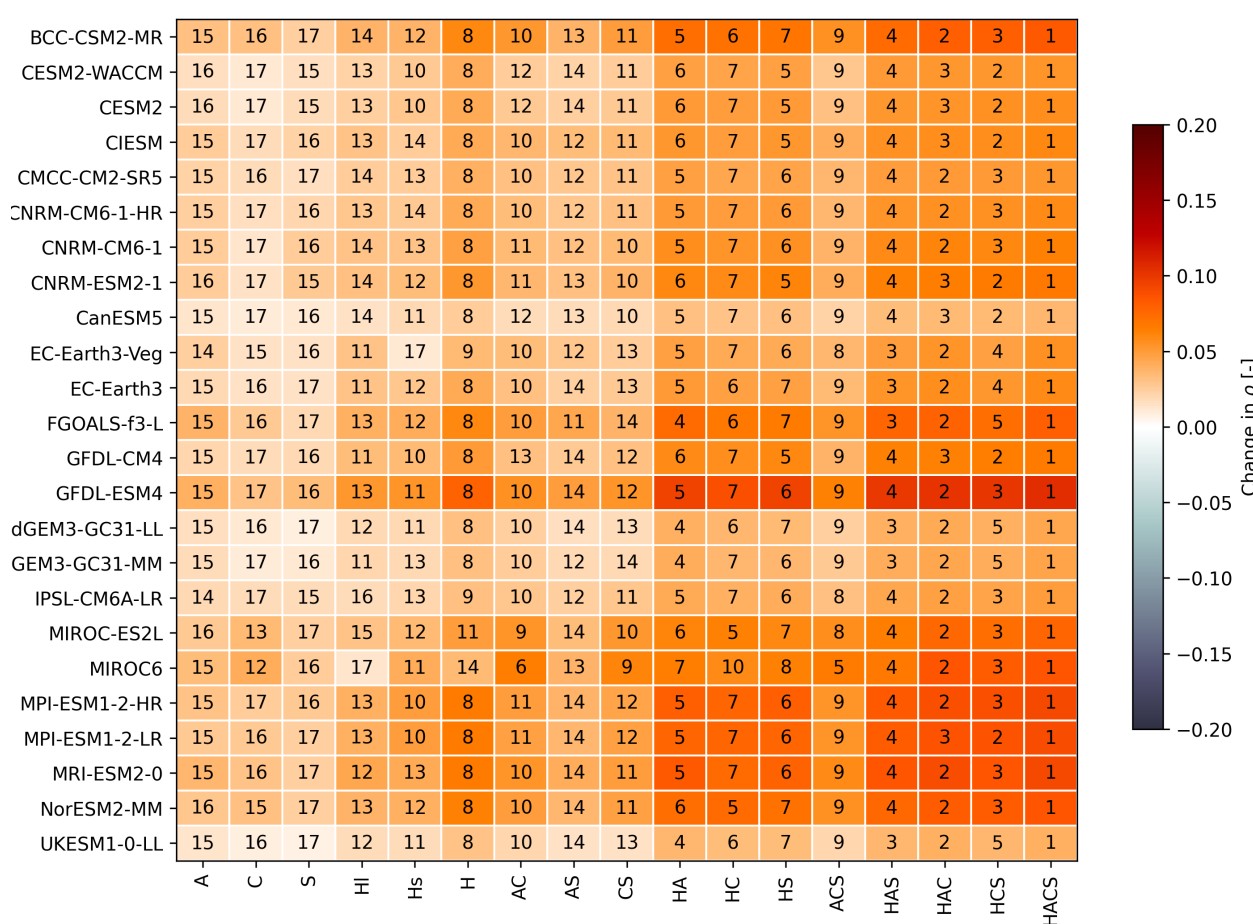

**Figure E1.** Global Land (without Antarctica) performances of the physical model trained using different predictor sets (columns) shown for all CMIP6 models (rows). Acronyms of the predictor sets (x-axis tick labels) can be referred back to in Table 1. Pearson Correlations calculated over all months between test runs and harmonic model test results augmented by the physical model's predictions of residual variability are considered. Here we show changes in correlations relative to those obtained when augmenting using only $T_{yr}$ and month values as predictors. Numbers indicate the ranking of each predictor set, where 1 is the best performing.





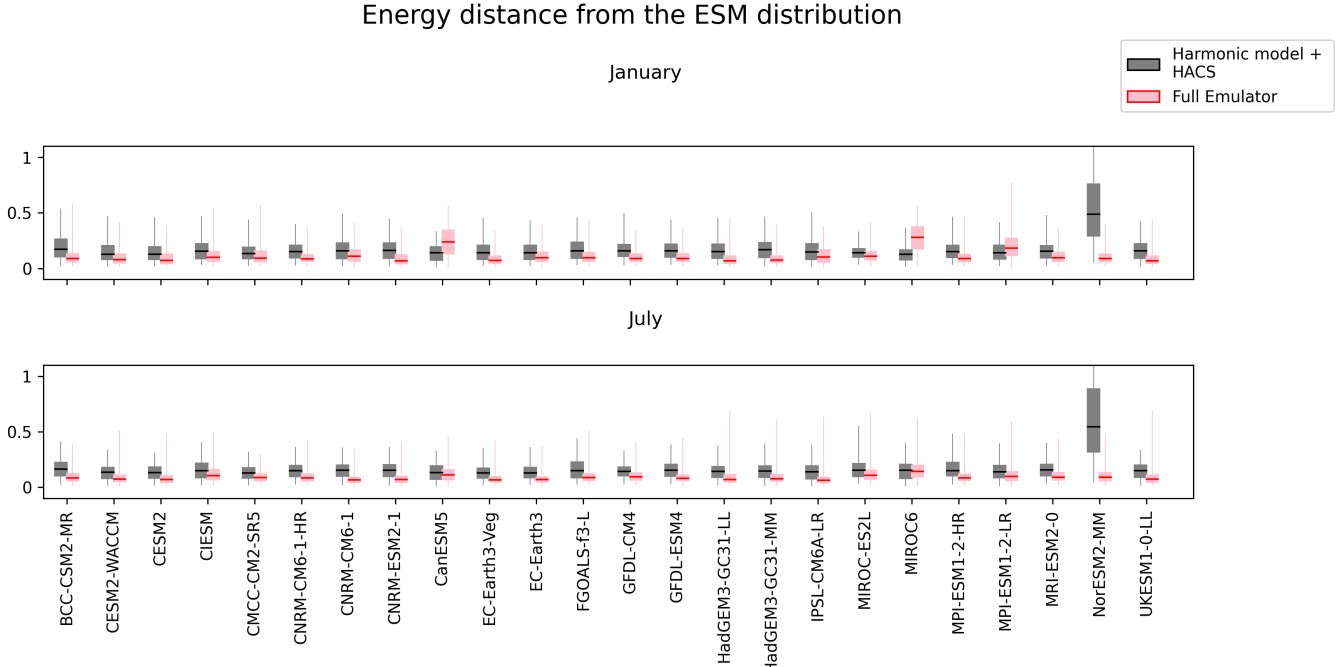

**Figure E2.** Comparison of the performance of the harmonic model + physical HACS model to that of the full emulator for January and July of all CMIP6 models. The energy distance from the actual model test runs is considered, where 0 indicates the best performance.

*Author contributions.* QL, CFS and SIS identified the need to extend MESMER's framework by a monthly downscaling module. SN designed the monthly downscaling module with support and guidance from QL and LB. SN led the analysis and drafted the text with help from 405 QL in developing the storyline. All authors contributed to interpreting results and streamlining the text.

*Competing interests.* The authors declare that they have no conflict of interest.

*Acknowledgements.* We acknowledge that this study was conducted as part of the LAMACLIMA project, receiving funding from the German Federal Ministry of Education and Research (BMBF) and the German Aerospace Center (DLR) as part of AXIS, an ERANET initiated by JPI Climate (grant no. 01LS1905A), with co-funding from the European Union (grant no. 776608). SIS acknowledges partial support from 410 the European Research Council (ERC) through the Proof-of-Concept Project MESMER-X (Project number: 964013). We furthermore thank Lukas Gudmundsson, Joel Zeder and Christoph Frei for their invaluable statistical insight into the development and analysis of the emulator modules. Finally, we thank the climate modelling groups listed in Table A1 for producing and making available the CMIP6 model outputs, without which there would be no training material for the emulator.





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
