# Peer review of "MESMER-M: an Earth System Model emulator for spatially resolved monthly temperature"

_Earth System Dynamics, 2021_

## Author Response (AR1)

Shruti Nath
Climate Analytics
Ritterstrasse 3, 10969, Berlin, Germany
Phone: +49 (0) 30 259 22 95 44
E-mail: shruti.nath@climateanalytics.org

[Figure]

Earth System Dynamics editorial board

Berlin, 03 February 2022

**MESMER-M: an Earth System Model emulator for spatially resolved monthly temperature**

Dear Dr. Messori,

Please find enclosed the revised version of our manuscript and a version with tracked changes. Directly attached with this letter are the point-by-point answers to reviewers we had posted in the interactive discussion. We found the reviewers suggestions and comments extremely helpful and having considered them, the following main changes were made to the manuscript:

1. A new, more elaborate schematic of the framework followed by MESMER-M has been provided with better visual illustrations. Additionally, the method section of MESMER-M (section 3.2) has been revised so as to be more clear e.g. by providing explicit equations for the Yeo-Johnson power transformation used.

2. We had initially calculated lag-1 autocorrelations for the AR(1) process and only applied the Yeo-Johnson power transformation on the resulting noise terms of the AR(1) process. This has been changed such that the Yeo-Johnson transformation is applied before the whole AR(1) process. We also make this more explicit in section 3.2.2 as well as the Figure 1 schematic describing MESMER-M's framework

3. We provide more discussion on the choice of train-test split in section 2 and justify its usage relative to other potential methods.

4. More discussion and consideration is provided towards modes of climate variability and outlook on how to possibly incorporate them in future.

Please refer directly to the revised manuscript for all additional changes we implemented based on the reviews. We are confident that the revisions increased the readability and depth of the manuscript.

Yours sincerely,

Shruti Nath

**(on behalf of all co-authors)**

**Referee 1**

The authors extend an annual temperature ESM emulator to simulate monthly temperatures that account for short timescale temporal correlation, spatial correlation, and intra-annual changes in distribution (e.g., variability and skewness). I find the paper to be overall well-written and a good contribution to the literature on ESM emulation, and therefore of interest to ESD readers, although there are a number of improvements that I believe could be made to both the content and clarity of presentation. Comments and suggestions follow:

We thank the Referee for their thorough review of the manuscript and the overall positive feedback. We found the insight provided - particularly on the technical parts as well as model verification (e.g. on the power spectra analysis) - extremely helpful and feel like the suggestions strengthened the paper in terms of future readers' understanding as well as robustness in methods. We tried our best to respond to the comments as below.

High-level Comments:

1. I typically think of an emulator as something that can be used to produce synthetic simulations under a new scenario that the ESM model has not been run under (or, alternatively, a new parameterization). The meaning here, by contrast, is something that can be used to generate additional synthetic simulations from the same scenario that the ESM was originally run under (SSP5-8.5 in your case), in order to better explore internal variability. That is fine, but I think it would be important to emphasize that you should not expect this emulator to perform as well under scenarios that produce very different GMST responses compared to the SSP5-8.5 scenario. That is, I think the pattern scaling model in Equation (1) is likely reasonable when applied to a single scenario, but would perform poorly if you tried to use the same pattern across very different scenarios.

Emulators have many avenues of application from exploration of model parameterisations to new scenario synthesis. In this study we indeed mainly focus on the applicability of MESMER-M to represent uncertainty in climate model projections resulting from internal variability, which is an irreducible model uncertainty and thus benefits from the significant reduction in computational costs provided by emulation. We chose to only use the extreme climate scenario, SSP5-8.5, so as to demonstrate the applicability of the MESMER-M approach on the extreme end of climate response types.This however does not dismiss the applicability to more moderate climate response types displayed within other climate scenarios. From preliminary testing, we believe that the yearly to monthly temperature downscaling relationship will be to the most part scenario independent- although this is still to be fully investigated. The pattern scaling equation from Equation (1) has also been demonstrated to work across climate scenarios (see Beusch et al. (2021) (in review): https://gmd.copernicus.org/preprints/gmd-2021-252/) as long as a wider scenario space is given as training material.

To clarify this we will add our reasoning on focussing on SSP 5-8.5 in L59-60 as well as add some discussion points on the inter-scenario applicability of MESMER done

according to Beusch et al. (2021) in the Conclusion and Outlook section highlighting the need for a representative training sample if inter-scenario exploration is pursued.

2. Your emulator allows for changes in variability and skewness within a year, but not across years (unless I misunderstand). There is evidence in model runs of changes in interannual variability over time, as well as changes in other aspects of the temperature distribution (like skewness and tail behavior). Did you look into this at all?

We did look at inter-annual changes in skewness of monthly temperature distributions, which led us to include the yearly temperature dependent lambda parameter in the power transformer. Some discussion points within the text that arose from these inter-annual changes in temperature distributional properties can be seen in:

- L229-231: lambda values >1 show a more positive shift in skewness of Northern-Hemispheric January temperature values (in line with other work e.g. Brodsky et al. 2020: https://doi.org/10.1038/s41561-020-0576-3, 2020)
- L232-234: The ice-free Arctic summer lead to a significant shift in skewness within Arctic July temperatures leading to a lambda parameter >1

To furthermore investigate whether the emulator was able to capture such changes in distributional properties, we split the investigation of quantile deviations into the period 1870-2000 and 2000-2100. In this manner we show how well the emulator captures inter-annual changes in distributional properties for 2 periods experiencing different magnitudes of global warming. To be more specific, the quantile deviation calculations take into account the positioning of the actual ESM run within the emulated temperature distribution at each year, thus if the emulator was not able to represent the interannual changes in skewness we would see significant quantile deviations in the period 2000-2100 as compared to those in 1870-2000. Given that the quantile deviation values between the 2 periods correspond reasonably well, we believe that the emulator's representation of such interannual changes in distributional properties is sufficient. We can elaborate on the choice to separate the quantile deviations into the 2 periods in section 3.3.3 (i.e. it enables investigation of interannual changes) as well as mention the correspondence of the 2 period's deviations and what that entails in section 5.3.

Specific Comments:

1. Equation (1), how is the smoothed GMST calculated?

The smoothed GMST is calculated using a locally weighted scatterplot smoothing (LOWESS), details of which can be found in the Beusch et al. (2020) paper.

2. Equation (2), the notation f_s(T_{s,y}) is confusing: you should indicate that this is also a function of month (m).

We agree that this is confusing and will add m to the equation such that it is f_s(T_{s,y},m)

3. L101, isn't the maximum value of n = 6 (not n = 8)?

We performed the BIC till n=8, however n greater than 6 was never chosen. We thus show an upper limit of 6 in the top panel's colorbar for Figure 2. For sake of simplicity we can however change n = 6 in L101.

3. L101, when you say that you use the BIC to determine the number of harmonic terms, what likelihood function are you using? Is this just the BIC assuming independent Gaussian noise (rather than your full model that accounts for spatiotemporal correlation and non-Gaussian behavior)?

The mean response module focuses purely on the deterministic part of the monthly temperature response to evolving yearly temperatures. We therefore assume independence from the Gaussian noise when fitting for this part. Our method in using the BIC for choosing the order of the harmonic model follows a brute force approach where we calculate the BIC for the harmonic model fitted (using Ordinary Least Squares Regression) at each value of n and choose the value of n with the lowest BIC score.

4. Equation (4) specifies a deterministic relationship; I believe you are missing a noise term and you need to specify its distribution.

We intentionally left out the noise term in Equation 4 as we are representing the time-correlation component (i.e. $\eta_{m,s,y}^{time}$) of the AR(1) process that we use to represent overall variability ($\eta_{m,s,y}$). The noise term is represented separately within the space component i.e. $\eta_{m,s,y}^{space}$ (equation 6). The complete $\eta_{m,s,y}$ is then the addition of the time and space components (combine equation 4 and equation 6) consisting of both lag-1 autocorrelations and noise terms. To make this clearer we could change L107 to read as

"Similar to previous MESMER developments, we represent the residual variability using an AR process. To produce spatio-temporally correlated noise terms, we delegate the noise component of the AR process as being space-dependent leaving the deterministic component as time-dependent, as shown in equation 3 (Beusch et al., 2020)."

And L116 to read as

"Following this, the noise component of the AR process, represented as $\eta_{m,s,y}^{space}$, derives its distribution based on spatial correlation structures and month-dependent skewness, expected to be non-stationary with regard to yearly temperatures..."

5. L118, it is not true that inferring correlation requires the data to be Gaussian. You could simply rephrase this to something like "We use a trans-Gaussian process relying on power transformations to account for the fact that temperatures may be non-Gaussian."

We think this is a good suggestion and will change L118 accordingly.

6. L121, I'd recommend that you define what the Yeo-Johnson transformations are, because I expect many readers will be unfamiliar with this.

We accept this suggestion and will add the equation for the Yeo-Johnson transformation before equation (5).

7. L121, I don't understand how the power transformations are being applied here. The typical approach would be to transform temperatures to approximate Gaussianity, then apply the additive model that you are using, then back-transform to the original scale. Is that what you are doing? If so, then I think the equation given in Figure 1 is incorrect. If not, can you please clarify where the transformation is happening and why you are using the approach that you are using (which would seem nonstandard to me)?

We initially sampled spatially correlated terms from the transformed Gaussian space and then back-transformed them independently within ($eta\_\{m,s,y\}^\{space\}$) before adding ($eta\_\{m,s,y\}^\{time\}$). This design choice had seemed most appropriate as we wanted the power transformer to derive the cleanest skewness to yearly temperature relationship. From further thought and given that the AR(1) process requires unconditional mean and variance in the sampling space, we decided to start with a power transformation before employing the additive models. Changes will be made in the text as required to clarify this.

8. L127, again what likelihood are you using when you say that you fit using maximum likelihood? Are you using the full likelihood that accounts for spatiotemporal dependence, or the likelihood that would assume independence?

We are using a likelihood that assumes independence. Future developments could look into one that accounts for spatiotemporal dependence however this was not focused on in this study.

9. L130-134, please clarify what the Gaspari-Cohn covariance function is and explain why you chose that covariance function.

The Gaspari-Cohn function was used in previous MESMER developments (see Beusch et al. 2020, equation 8) and allows for exponentially vanishing correlations with distance such that anisotropy of spatial cross-correlations on regional scales is still retained. We chose it for consistency within MESMER as well as the aforementioned property. Since it was already elaborated on in the previous MESMER paper we do not go into its specifics. To make this clearer however we can add to L134

"... localized by point-wise multiplication with the smooth Gaspari Cohn correlation function (Gaspari and Cohn, 1999) which has exponentially vanishing correlations with distance $r\_m$."

10. L155, I don't think it is a good idea to only look at the frequencies corresponding to the top 50 highest power. This will cause you to focus on low-frequency variability, but the total variability is the integral of the whole spectrum and there are more high frequencies than low frequencies (so it will typically be the case in my experience that the integral over the higher frequencies is comparable to or larger than the integral over lower frequencies, even though the power is lower at higher frequencies). Put another

way, short timescale variability typically dominates, even though the spectrum is higher at lower frequencies. As such, if you get the high-frequency variability wrong, this can have very important consequences even if the power at individual high frequencies is small.

This is a good point and we will consider switching to looking at the representation of the top 50 highest frequencies instead.

11. Section 3.3.3, I don't understand what is being done here. Are you averaging within the region and then taking the 5th, 50th, and 95th percentiles over time? Or are you calculating those quantiles across individual gridcells within the region?

We take the regionally averaged temperature time series over which the percentiles are then calculated over time. We could elaborate on this by modifying L163 to read as

"The emulator is evaluated for its representation of regionally averaged monthly temepratures of all 26 SREX regions (Seneviratne et al., 2012) at each individual month (see Appendix B for details on SREX regions)."

12. L237-240, I don't understand why the spatial model should depend on the number of runs in the ensemble. You should be able to better estimate the spatial model with more runs, but the spatial model shouldn't change. Am I misunderstanding what you are saying here?

L237-240 point out that the localisation radii, r_m, will be larger with a larger number of ensemble runs. This follows from r_m being estimated using the leave-one-out approach, such that models with larger ensemble runs will have more years within their training set and thus more options to "leave-out" and calculate the likelihood over. Such encourages the selection of higher localisation radii as there is more evidence towards the spatial correlations existing up to higher distances. In contrast, models with fewer ensemble runs will prefer the simpler option and choose lower localisation radii due to limited options available to "leave-out" and calculate the likelihood over, while lower values are bounded by zero. We acknowledge that the determining factor for the localisation radius chosen is the model itself, however we mainly wanted to indicate that in coming closer to a localisation radius true to the model properties, the number of ensemble members could also be a bottleneck i.e. with less training material the model will settle for the simpler option.

13. Figure 1, what is going on with the gridcells showing very high autocorrelation values (and, likewise, with those showing very negative autocorrelation values)? Are these locations where the mean model is not performing well?

We interpret places where there are higher magnitudes of autocorrelations as areas where there are other month-to-month processes that are not fully covered by the mean response model. This is not necessarily a shortcoming on the side of the mean model but may just mean that there are additional month-to-month correlations arising from

processes that do not directly follow from changes in yearly temperatures but for instance, are even amplified/dampened.

14. Figure 3, It looks to me like the emulator runs are more variable than the ESM runs. Is that the case, or just an artifact of how the plot is displayed? Can you give a direct comparison of the variability from the two? Also, there are some visible problems with the mean model especially for MPI-ESM1-2-LR in WAF that are not discussed in the text.

The figures may be misleading as the number of emulations are much higher than number of ESM runs and we will correct this so as to show the emulations using a 2D histogram.

The overall variability representation can be deduced from the quantile deviation plots, where we have a typical problem of underdispesivity within the emulator due to localisation of spatial covariations (See Beusch et al. (2020) where this problem was explored too).

Thanks for raising the mismatch between the MPI-ESM1-2-LR runs and their emulations in WAF. We had investigated this (not discussed in the paper), and strongly suspect that this is related to the dynamic vegetation scheme in this model. Tree fraction strongly decreases (by ~30%) in that model between 2030 and 2050, and we concluded that the resulting climate changes driven by biogeophysical changes in albedo and surface heat fluxes dominate the ESM runs during this period. We will add a short discussion on this at the end of this section.

15. Figure 4, It looks to me like the emulator is under-representing skewness particularly in WAF for regionally averaged temperatures. I wonder if this is the result of a deficiency in the model for spatial correlation, or if rather the power transformation you are using is not sufficient for transforming to Gaussianity.

We acknowledge the problem in under-representing skewness and believe it arises from limited degrees of freedom (i.e. the power transformation solely relying on yearly temperature) to capture skewness. We discuss this in L305-310, later on in the paper,:

"To be exact, even though changes in skewness in local monthly temperature residuals with evolving local yearly temperatures is incorporated, such cannot completely represent delayed or compound effects, otherwise brought about by secondary/external feedbacks (Shinoda,2001; Potopová et al., 2016), which more largely affect extreme quantiles (Huybers et al., 2014; Guirguis et al., 2018; Loikith and Neelin, 2019)."

We can perhaps move this discussion up to after Figure 4 as it may be most relevant there given the strong visual evidence. Since the change in monthly temperatures over WAF may be dominated by other effects such as those arising from changes in land cover (see discussion in comment above), this is furthermore a strong example of the shortcomings of solely using yearly temperatures as input for inferring changes in distributional properties (this can also be added to the lines above).

Having noted these shortcomings, we have included ideas for further emulator developments within the Conclusion and Outlook section (e.g. develop a LCLM mean sub-module), but to directly link it to the problems seen in Figure 3 and 4 we can preface the suggestions with a reiteration of the shortcomings in the mean response and variability module due to yearly temperature being the sole input.

16. L268 - 276 and Figure 6. I don't understand whether you are saying that you under-estimate the spectrum when the power is high, or that you under-estimate the spectrum for high frequencies. Can you please clarify? I also don't understand how to read the inset example in Figure 6, which doesn't seem to give information about frequency. Rather than plotting the power in one vs. the power in the other, I'd recommend that you plot the ratio of spectra vs. frequency.

We mean that we underestimate the spectrum when the power is high. As a response to comment 10 however, we will in future look at the spectrum of high frequencies and will clarify this within the figure caption. The suggestion to plot ratio of spectra vs. frequency is also a good idea which we will implement.

**Referee 2**

General comments

Climate model emulators are becoming more and more useful in assessing climate change through representing complex Earth system model (ESM) behavior and combining many different ESM simulations with other multiple evidence. The MESMER approach, being developed by the authors, is unique as an emulator for generating spatially-resolved forced and unforced climate realizations based on multi-ESM, initial-condition ensemble simulations. This paper describes the newly developed MESMER-M module for monthly downscaling, which is expected to further expand the emulator's applications.

Although the current manuscript adequately describes the structure and performance of MESMER-M, there is room for improvement as follows, which should be appropriately revised for publication.

We thank the Referee for their thorough review of the manuscript and the overall positive feedback. Several good points were made about the need to acknowledge major variability modes such as ENSO and changes in atmospheric circulation patterns. We also found some suggestions in tackling these quite helpful and integral in strengthening our discussion as well as the climatological perspective of the paper. We hope our response addresses these aspects sufficiently. We furthermore value the comments made for better clarification of the methods section and hope that changes made enable better understanding of the methods for the broader non-technical audience.

(1) The calibration and verification results indicate marked dependency on the number of ensemble members, which may raise concerns about the robustness of the methods.

Such dependency appears in most key parameters and performance aspects: the order of the autoregressive process in the temporal variability module (L218 and L239), the stationarity of the shape parameter λ in the spatial variability module (Appendix D), the scale parameter $r_m$ for the localization in the spatial variability module (L237, L283, Figure 7), and the bench mark test with GBR (L338). Although each of these results is explained in terms of the amount of training runs as input information, it is not much convincing. Implications for the robustness of the methods and a possible guideline of an appropriate size of the training runs should be discussed thoroughly.

General MESMER fitting recommends the training of the emulator on all available ensemble members (see Beusch et al. (2020)). While that means that for some models the emulator will have more training material, it also follows the philosophy of feeding as much information as possible into each model-specific emulator so as to generate the best possible "super-ensemble". There are other approaches in getting the best training set size, such as that employed by Castruccio et al. (2019) so as to balance the stability in the inference (represented for example by variability) of the emulator, and benefits for reduction in computational costs. Such approaches however require the presence of a large ensemble and would mean that we would be constrained to demonstrating MESMER-M's performance on a smaller subset of the available CMIP6 models. We thus settle for a train-test split of approximately 70-30 as this provides some stability in inference whilst maintaining low training time and leaving aside samples for validation. It should be noted that, even though we explain model calibration results as dependent on training runs, we do not wish to discount such calibration parameters being dependent on the model itself (as referred to in previous MESMER fittings and mentioned in our Conclusion and Outlook, the calibration parameters represent unique model IDs). We will try edit the text so as to make this clearer as well as add a discussion point on the need to choose training set sizes such that stability in inference is stable whilst computational costs are kept low (as according to Castruccio et al. (2019)).

(2) The modules and the calibration and verification results lack interpretation from a climatological point of view. Although the seasonality associated with snow cover is frequently mentioned, this is just one aspect. It is necessary to describe and discuss the validity of the modules from the aspect of major variability modes, such as monsoon, ENSO, and AO.

Deviations from the mean seasonal cycle may not necessarily be biophysical feedbacks, as assumed in L184-185. Internal climate variability leading to some deviations, such as jet meandering and blocking associated with the strength of the polar vortex, is hardly regarded as a biophysical feedback. It is not much convincing that the difference between summer and winter in the mean response verification (5.1) is explained by the snow-albedo feedback only. In the regional-scale verification (5.3), although the increasing deviations in July between the ESMs and the emulator (L294-) is worth being

noted, it may also need to be described based on specific natural variability, rather than regarding the tendency as abstract secondary, non-linear responses.

See also the specific comment on the conclusion and outlook below.

Major variability modes such as monsoon, ENSO and AO can indeed drive changes in interannual variability and we acknowledge their relevance in considering monthly temperature responses to GHG induced warming. We propose adding some discussion points within the Conclusion and Outlook on how to tackle their representation as further elaborated in response to the specific comments. We furthermore can modify L184-185 to clarify that we do not expect biophysical variables as the sole explanatory variables and that other atmospheric circulation processes may also play a role. We had only focussed on biophysical variables in investigating residual variability, having considered their advantages and relevance towards the primary purpose for which we employed them (i.e. model benchmarking) as follows:

1) These biophysical variables are readily available output within CMIP6 models, providing easier access as explanatory variables within the Gradient Boosting Regressor (GBR) model, with less degrees of uncertainty vs having to deduce jet meanderings, atmospheric blockings, ENSO, AO from for instance SSTs or soil moisture.
2) Given that within our monthly emulations we are also interested in intra-annual variability, we expect biophysical variables to most effectively "kill two birds with one stone" providing information on the drivers behind both changes in intra and inter annual variabilities that the GBR can effectively sift through.
3) Atmospheric circulation patterns can be expected to leave their fingerprints on for instance latent & sensible heat fluxes, such that even though we cannot directly attribute temperature variability to atmospheric circulation patterns we do still have a good representation of temperature variability against which to benchmark MESMER-M.

Unfortunately, given that we mainly focus on the technical aspects of MESMER-M we intentionally chose not to delve into a full climatological investigation as this would add an extra layer of complexity that is out of scope of our aim: providing simple statistical emulations of monthly temperatures from yearly temperatures. Furthermore, following studies such as Schwingschackl et al. (2018) we saw the added benefit of pursuing a biophysical based GBR representation of temperature variability as this allows extraction of biophysical IDs, which are of relevance to the Earth System Modelling community (e.g. for model benchmarking as proposed within the Conclusion and Outlook). Our analysis within L294- mainly focuses on pinpointing shortcomings within the emulator performance so as to highlight caveats of its usage, and in such we did not fully elaborate on the underlying processes but refer to papers where they have been thoroughly investigated. We do agree that it would be a better idea however, to add to the aforementioned lines that it is not just biophysical feedbacks but also atmospheric drivers at play.

(3) From the standpoint of potential users of the series of MESMER modules, who are not necessarily familiar with technical details, it is recommended to devise some descriptions for better understanding.

For example, in Figure 1, adding X-Y plots illustrating a typical seasonal cycle and its variability and skewness would help understand the local variability module. Visual materials would be useful for making sense of technical concepts like the multivariate trans-Gaussian process and the Gaspari-Cohn function.

In terms of consistency between MESMER and MESMER-M, it may also be useful to verify whether the annual average of each element of MESMER-M is consistent with corresponding elements of MESMER.

MESMER-M is planned to be made open-source with its own Github page, we agree that there it will be of relevance to provide visual representations when describing each module. We will furthermore experiment with adding some in text grid point X-Y snapshots to Figure 1, for ease of readers understanding as well as visualisation of the power transformation process in section 3.2.2.

Specific comments

L59. A brief explanation about limiting the scenarios to high emission SSP5-8.5 and applicability to low emission scenarios would be helpful.

Our rationale between training on SSP5-8.5 was to get the extreme end of monthly temperature response to yearly temperatures. In general, we would expect the yearly to monthly temperature downscaling relationship to be relatively scenario independent such that training on the extreme SSP5-8.5 scenario allows a rough capturing and validation over the whole spectrum of monthly temperature response types. To explore inter-scenario applicability however, the emulator should of course be trained across all scenarios. To elaborate on this we could add:

"... so as to first explore the emulator's applicability to the extreme end of GHG induced warming"

L61-62. The 70-30 train-test split is not consistent with actual split shown in Table A1. It appears that the 70-30 ratio is rather exceptional, and that some models with a large number of members have 50-50.

We roughly followed a 70-30 split, however for MIROC6 and CanESM5 a 50-50 split was done, as training on more than 10 ensemble members led to significant training time with no real gain in model performance. We can modify L61-62 to read as

"...is done in a roughly 70-30 manner, and for models with more than 20 ensemble members a 50-50 manner so as to maintain a good balance between training time and model performance."

L65-66. It should be clarified how the anomalies are calculated, i.e., whether they are deviations from the annual climatological mean or from the monthly climatological mean.

To make this clearer we can modify L65-66 to read as

"... annual climatological mean over the reference period of 1870-1899."

L82, 85. The use of the term "forcing" can be a bit confusing. As "other external forcings" imply an underlying primary forcing, "a certain forcing" may be better rephrased in a specific way. Changes in land cover can be anthropogenically forced or induced by climate change and variability. A more specific wording may be necessary to avoid misunderstanding.

We accept this and will change the use of "other external forcings" accordingly.

L90. The term "monthly cycle's mean response" is a bit, confusing considering the subsequent "seasonal cycle". "Monthly mean response" may communicate its intention without ambiguity.

We accept this and will replace "monthly cycle's mean response" accordingly.

L96-97. It appears that the need for high-order harmonic terms is not convincing. My understanding is such that up to the second order term representing a bi-modal cycle is enough for the mean monthly response. Are the month-to-month correlations, which is the case for some natural variability modes, out of scope for the mean response?

More flexibility in the order of harmonics chosen was allowed, as even though only 2 orders would be needed for bimodal representation, this could be too simple in terms of representing complex changes in the amplitude of the seasonal cycle with evolving yearly temperatures. In such, we deliberately allowed more degrees of freedom for order of harmonics, whilst using the BIC to ensure model complexity still made sense with respect to accuracy in mean response representation.

L111. Check "time-dependent and space-dependent components" is correct wording. They are functions in terms of month, space, and year. Maybe, tempral-variability and spatial-variability components.

This is a good suggestion, and we will implement the changes accordingly.

L112-114. Is this an appropriate explanation for adopting a autoregressive order-one process model? AR(1) may be suitable when the autocorrelation function of the stochastic process has significant components up to lag three or so.

In this case, as the variability module is month specific, the autocorrelation function for a month is built independent of that of other months. Hence, we would expect that accounting for lag-1 autocorrelations would only represent the month-to-month

correlations for subsequent months and not more i.e. up to lag one only. It should be noted that here we are also mainly referring to the deterministic part of the AR(1) process.

L130-136. The purpose of localization should be explicitly stated, which would be helpful for the relevant issue described in the paragraph starting L302.

The need for localisation if elaborated in the Beusch et al. 2020 MESMER paper and hence we chose not to repeat the reasoning. To clarify this, we can modify L133 to read as:

"...and covariance matrix, Sigma_{nu_m}. Similar to previous MESMER fittings Sigma_{nu_m} is rank deficient (Beusch et al. 2020), and is thus localized by point-wise multiplication with the smooth Gaspari Cohn correlation function (Gaspari and Cohn, 1999) which has exponentially vanishing correlations with distance r_m."

L140. In equation (7), is there a case where the magnitude of $\gamma_1$ is greater than 1? Figure 2 shows that some models have means close to ±1.

The absolute value of γ1 is constrained to less than or equal to 1 during fitting of equation (4). We can specify this in L115 as:

"... where  gamma_{1,m,s} is between -1 and 1."

L145-146. Specify whether area weighting is processed or not.

To clarify the above we can add:

"... across the whole globe for each month with each grid point weighted equally."

L168-170. This quantile comparison procedure is unclear.

We propose providing a numbered step-by-step procedure of this within this section.

L194-197. This sentence is complicated and should be clarified more.

We propose dividing the sentence as follows:

"Pearson Correlations over all months, between ESM test runs and harmonic model test results augmented by biophysical variable, T_{yr} and month based physical model predictions are calculated. As a measure of performance the aforementioned correlation values are given relative to those obtained when augmenting using only T_{yr} and month based physical model predictions."

L207. The symbols in Equation (8) are not fully described. Instead of this equation, the integral of difference between two CDFs would be more understandable as a definition of the energy distance.

The energy distance is not exactly the same as the difference between 2 CDFs but was proven to be equivalent to the square root of twice the distance between 2 CDFs (see Székely: The energy of statistical samples (2002)). This relationship however is dimension dependent and to be consistent with the actual method used to calculate the energy distance as well as emphasise the non-parametricity of it, we chose to use the formula as given in equation (8).

L224-225. Is there anything to be added? Readers would be curious about what kind of characteristics of the two models result in such outlier results.

Unfortunately we could not think of any obvious reason for such model behaviour. We additionally refrained from going into too detailed analysis of each model as that would require a more in-depth analysis of the model documentation itself, which is beyond the scope of this study.

L233. The description about the equatorial region appears to be limited to January, if so, it should be stated as such.

We can rephrase L233 to read as:

"Around the equator (-5° to 5°) we see lambda^{tilde}_{m,s} values consistently higher than 1 especially in the month of July, with INM-CM5-8 and INM-CM5-0 displaying significantly high values.

L248-254. Readers would be curious about the MIROC6 results, in which relatively many ESM points appear outside of the emulator range in WAF. Is there anything to be mentioned in this regard?

MIROC6 in general shows higher interannual variability in WAF as compared to other ESMs for which the addition of variability terms tends to be under dispersive. We are not sure about the exact causes of this but suspect an influence of cloud feedbacks that do not directly relate to changes in yearly temperatures. Since we focus on more technical aspects of the emulator in this paper however, we refrain from going into climatological conjectures of individual regions and ESMs.

L258-260. The authors mention different timing of changes in snow cover among ensemble members for lower correlations in April and October, but this is not the case for several ESMs that have only one training run and zero test run.

We agree with this and realise that the low correlations also arise from inter-annual differences within each run in the timing of snow cover increase and decrease, hence we will modify these lines to read as:

"Such low correlations could result from the inter-annual spread in the timing of snow cover decrease and increase, such that the mean response extracted does not always match individual years."

L275. Meaning of "such higher power spectra" appears unclear.

For more clarity we will replace "such higher power spectra" with "higher order temporal patterns".

L291. It looks mixed positive and negative rather than "generally low values."

To clarify we will replace "generally low values" with "generally low magnitudes."

L288-289. Is it correct for "deviations of the ESM training (and where available test) runs from the full emulations"? Figure 8 caption writes quantile deviations of the monthly emulated quantile from that of its ESM training and test runs although the embedded figure title is different. Which is the base for the deviation?

The base of the deviation is the emulated quantile. We will correct Figure 8's caption to read as:

"... quantile deviations (colour) of the ESM training run values from their monthly emulated quantiles…"

L311. Around here, it is hard to trace relations between descriptions in the text and corresponding parts in the figures. Even if the complexity of the figures is unavoidable to some extent, main points and their implications should be clearly indicated.

We see how this can be confusing and will replace L309-314- such that the main points are emphasised -as follows.

"For January over the time period of 1870-2000, lowest magnitudes in 5th and 95th quantile deviations is observed for Southern Hemispheric regions (e.g. AMZ, NEB, WSA, SSA) , along with slight overdispersivity (see the blue, respectively red values in the left, respectively right panels of Figure 8). Over the period of 2000-2100, this behaviour for January switches to Northern Hemispheric regions (e.g. CEU, ALA, ENA, WNA, TIB) and is mostly apparent for the 95th quantile, possibly due to a decrease (increase) in January variability in the Northern (Southern) Hemisphere with increasing yearly temperatures \cite{Holmes2016}. In contrast, over both the periods of 1870-2000 and 2000-2100, July consistently displays lowest magnitudes of 5th and 95th quantile deviations (with even slight overdispersivity) in Northern Hemispheric regions (e.g. WNA, ENA, NAS, WAS). Regional dependent overdispersivity indicates that the vanishing of spatial cross correlations with distance is contingent on the region itself, such that some regions have less spatial cross-correlations with distance than others. Since the spatial covariance matrix is localised with a single global number, the variability terms for such regions therefore account for more spatial cross-correlations than necessary leading to the observed overdispersivity."

We will furthermore add some discussion points on tackling improving the localisation of the spatial covariance matrix, for example by expressing $r_m$ as a function of latitude.

L341-344. Figure E2 shows an exceptionally bad performance of NorESM2-MM. Is there anything to be mentioned?

We are not sure why NorESM2-MM has such bad performance, it is possible that there is some key biophysical information lacking and the biophysical variables investigated within this study don't play much of a role in explaining the variability for this ESM. We do not go into detail about this however as it would be beyond the scope of this study.

L335-338. Regarding greater CDF distance in January, Figure E2 may not clearly indicate such distinctive difference between January and July. Grid lines in the plot space would be helpful for identifying the difference.

For bringing out the difference we will add grid lines to Figure E2

Figure E2 also shows that CanESM5 has greater distance in January as well as MIROC6 and MPI-ESM1-2-LR.

We are aware of this but do not mention it as we focus primarily on MIROC6 and MPI-ESM1-2-LR within section 5.4.

Conclusion and outlook. While the emphasis is on future developments that take into account biophysical variables, there remains the question of how similar the variability components by each ESM are to observations, although the latter is out of scope in the current manuscript. What needs to be focused on in this regard would include properly emulating how major variability modes, such as ENSO, modulate with warming, considering any dependence on ESMs. If a certain variability mode includes some memory effects associated with , for example, land soil moisture they may need to be modeled by higher-order autoregressive processes, and if the variability affects remote climate on a global scale through teleconnections, the localization of the spatial covariance structure may need to be improved. In any case, future developments will need to be described in a broader scope.

We acknowledge that the representation of higher order AR processes possibly arising from major variability modes are not modelled for. We had looked at AR processes up to order 3 but the coefficients' magnitudes decreased significantly after order 1 hence we opted for an AR(1) process (we could include a demonstration of this within the supplementary materials). We did also include soil moisture within our biophyiscal variable exploration, however it proved unimportant when latent heat flux was included and hence we chose not to go further with it. It could be of interest  to investigate the lag correlation between soil moisture and the temperature variability, however this seemed out of scope for us given that we wanted to emulate monthly temperatures from yearly temperatures only. We will however add a discussion about this, as well as link it as a possible method of pursuing representation of variability modes (e.g. ENSO) as possible future avenues in MESMER development.

We are aware of a monthly emulator developed by Mckinnon and Deser 2018 which deals with modes of ENSO, PDO and AMO, requiring SSTs as input. Methods of

including bits of their approach within MESMER-M were also considered, however we were limited by yearly temperature being the sole input. We can however add a point on this within the Conclusion and Outlook, highlighting the shortcoming of the representation of global-scale circulation patterns within MESMER-M, but also emphasising the need for combined emulator approaches as each emulator has its own strengths in representation.

In case of investigating global scale teleconnections, we had attempted a Principal Component Analysis based approach to including teleconnections. However, the added degree of freedom did not add much information as the main local monthly temperature response followed directly from local yearly temperatures and otherwise corresponded to local changes (for instance in land cover). Hence, disentangling the variability modes arising from teleconnections and their manifestations on a local level proved difficult and we decided to follow a simpler approach of generating spatio-temporally correlated residuals instead. While exhibiting the added value of being consistent with previous MESMER developments (Beusch et al., 2020), we hope to demonstrate in this manuscript that it achieves a good performance.

Acknowledgement. Refer to https://pcmdi.llnl.gov/CMIP6/TermsOfUse/TermsOfUse6-1.html to confirm whether acknowledging CMIP6 is appropriate, and requirement for citing CMIP6 model output.

The link indeed suggests an acknowledgement following language such as:

"We acknowledge the World Climate Research Programme, which, through its Working Group on Coupled Modelling, coordinated and promoted CMIP6. We thank the climate modeling groups for producing and making available their model output, the Earth System Grid Federation (ESGF) for archiving the data and providing access, and the multiple funding agencies who support CMIP6 and ESGF."

We will also double check the citation requirements.

Figure 3 and 4. Units are missing. The figure legend implies that plotted data are three members for the ESMs, the three patterns for the sum of the yearly and seasonal cycle, and 50 realizations for each of the three patterns for the emulation, but should be fully described in the caption text. The second sentence of the caption text should be clarified about whether the reference temperature is monthly or yearly.

This is a good point and we will add this to the caption test. We will clarify that a yearly reference temperature is used as well.

Figure 6. Although the text in 5.2 (L271) implies that the box plots in Figure 6 show distributions across different realizations by the emulator, the figure caption should be fully described about how different bands (50 elements), different emulator realizations (50 members), and different training and test runs (ESM-dependent ensemble number)

are processed to represent the data distribution. Also, if the whiskers in this figure (Figure 7 as well) are drawn as in Figure 2, indicating a min-max range, explicitly state so, otherwise describe it accordingly.

This is a good point and we will include the different bands and emulator realisations used as well as the range represented by the whiskers within the caption text.

Figure 8. It would be good to have a guide so that readers easily identify representative geographical zones corresponding to the individual regions alighted on the horizontal axis.

We were afraid that providing a visual guide would take up more space and add an extra layer of complexity to the figure. However, we will explicitly refer to Figure B1 to provide visual guidance.

Editorial comments

L70. Inline Mathematical symbols should be italic.

We will change this accordingly

L73, L95. Indent is unnecessary.

Indent will be removed

L105. Section 3.2.1, not section 4.1.1, but this indication within the parenthesis is redundant.

This reference will be removed

L110, L115, etc. Add comma to the end of the preceding expression and lowercase "Where".

This change will be implemented accordingly

L136-137. "based on", not "based off."

This change will be implemented accordingly

L169, L175, L202, etc. Long dashes (em dashes) are not typesetted correctly.

We will correct this

L183. "properties of the monthly temperature response" (maybe "of" is missing)

We will add ''of"

L203, 204, 220. "added ontop" and "ontop of" may not be common wording.

We will change "added ontop" to "added" and "ontop of" to "additional to"

L224. HadGEM3-GC31-LL, not HadGeM3-GC31-LL. Put it after ACCESS-CM2 if alphabetically ordered.

This change will be made

L234. "this" in "The source of this" is unclear.

We will change "this" to "the aforementioned"

L237-238. In this context, "boreal winter", not just "winter". Pay attention to whether inappropriately referring to specific seasons in terms of the Northern Hemisphere throughout the manuscript.

We will add "boreal" to clarify this

L241. "four selected ESMs", instead of "a select 4 ESMs." See the journal's English guidelines for numbers, and, if needed, spell out numerals less than 10 throughout the manuscript.

Change will be implemented accordingly

L244. It is unclear "all ESMs" are the four selected ESMs or all the ESMs used in this study.

We will change "all ESMs" to "the four selected ESMs"

L328-329. "latent and sensible heat fluxes", not "latent heat fluxes."

Change will be implemented accordingly

L163. Spell out SREX here. Also in Figure B1 caption.

SREX will be spelt out in the 2 places

Figure C1 and C2 captions (also, Figure D1 and D2 captions). For Figure C2, "Same as Figure C1, except for January" would be fine. Referring to the Benjamini/Hochberg correction may be required in the Figure C1 legend.

We will implement this suggestion

Figure E1. It might be better to replace X and Y axes for comparison with Figure 12. A more concise caption would be "Same as Figure 12, except that all CMIP6 models are shown for the global land."

We chose the X and Y axes for aesthetic purpose as otherwise the score numbers would have to be either squashed or rotated. We will implement the caption suggestion.

**Referee 3**

General summary: This paper extends the existing MESMER approach to include a monthly downscaling module, to enable the generation of large ensembles of spatially explicit monthly temperatures that are representative of ESM behaviors, which could be an useful tool for regional impact assessments. The paper is clearly written for the most part and contributes to the existing literature on ESM emulators. However, I have some concerns that I would like the authors to fully address.

We thank the Referee for their thorough review of the manuscript and the overall positive feedback. The general comments provided a useful compass on what parts of the paper were unclear and we hope that our proposed changes better clarify these parts as well as emphasise aspects of the text that are vital towards the overall analysis. The specific comments were also useful in strengthening the text for the broader audience. We hope that we addressed the comments sufficiently below.

General comments

(1) The training and verification results (such as those described in Lines 218-219, Lines 238-240) show non-trivial dependency on the number of ensemble members available for training, which raises concern not just for the robustness of this method, but also the usefulness of the method presented. If MESMER-M relies on a large number of ensemble members to get robust results, then it defeats its own purpose. A useful component to add into the paper is a sensitivity test to show what is the size of training runs needed to get a robust training result.

General MESMER fitting recommends the training of the emulator on all available ensemble members (see Beusch et al. (2020)). While that means that for some models the emulator will have more training material, it also follows the philosophy of feeding as much information as possible into each model-specific emulator so as to generate the best possible "super-ensemble", without judging the realism of the training sample. There are other approaches in getting the best training set size, such as that employed by Castruccio et al. (2019) so as to balance the stability in the inference (represented for example by variability) of the emulator, and benefits for reduction in computational costs. Such approaches however require the presence of a large ensemble and would mean that we would be constrained to demonstrating MESMER-M's performance on a smaller subset of the available CMIP6 models.

In this paper we wanted to show the ability of MESMER-M in representing monthly temperature distributions across all CMIP6 models without penalising models with smaller ensemble sizes. While the calibration results show some dependence on ensemble size, we should stress that this is simply a bottleneck and the model itself is the main driver of the calibration results (e.g. even with only one ensemble member MCM-UA-1-0 has a high localisation radius). It should also be stressed that the performance of MESMER-M is not dependent on ensemble size as for instance the pearson correlations within the mean response module as well as the quantile deviation magnitudes within the regional-scale verification are quite similar across all models. Furthermore, even though we attribute the performances of MIROC6 and MPI-ESM1-2-LR within the model benchmarking exercise to the ensemble size, this is not a universal feature as it only appears in January. We will try

make clearer that the ensemble size is not the sole determining factor within the text, as well as emphasise that the primary purpose of MESMER-M is to provide the best possible emulations based on training material available. It could be worth doing such sensitivity tests however it goes beyond the scope of this paper and we propose mentioning it in the Conclusion and Outlook section instead.

(2) The technical details need to be better described/clarified, for the potential users of MESMER-M to fully understand the approach taken, the assumptions made, and the procedure to carry out the training, calibration, validation, and generation of 'super-ensemble' using this method. Please see some of my specific comments below.

We see how the technical details may get a bit complicated and hope that the improvements made from the specific comments below as well as those from the other 2 reviewers improves this. We will furthermore experiment with visualisation of the methods by providing X-Y snapshots in Figure 1 as well as a schematic of the power transformer process in Section 3.2.2.

(3) My understanding is that the particular MESMER-M presented here can generate a 'super-ensemble' under the SSP5-8.5 scenario. Although the authors stated that 'MESMER offers the perspective to improve our understanding of the likelihood of future impacts under multiple scenarios', a different version of MESMER and MESMER-M has to be developed per scenario. If so, this needs to be clearly stated, and could the authors comment on how straightforward this process would be to expand this work under multiple scenarios.

It has already been demonstrated that one MESMER can be used to represent different climate scenarios, if it is trained across a representative sample of scenarios (See Beusch et al. (2021): https://gmd.copernicus.org/preprints/gmd-2021-252/). From preliminary testing, we expect the monthly to yearly temperature relationship to be fairly scenario independent as well. To clarify the process needed to expand MESMER-M to other scenarios we will elaborate on the existing expansion of MESMER as well as the need of a training set representative of all scenarios within the Conclusion and Outlook section.

(4) The fact that the emulator solely relies on annual temperature as input, and the assumption that other forcings have very little impact on the local monthly temperature response, makes the applicability of the monthly temperature probability distributions derived from the emulator limited. There should be more discussions around in what applications would the emulator results be particularly useful in Section 6.

We took this emulation exercise as a challenge in seeing how much of the monthly temperature response could be reproduced using yearly temperatures as the sole input and in the simplest manner possible. We do acknowledge that the assumption of little impact from other forcers limits the applicability of MESMER-M, particularly when other forcings have a strong impact on the monthly cycle (e.g. changes in tree cover due to deforestation in WAF for MPI-ESM1-2-LR see Figure 4 and refer to comments 14 and 15 for referee 1) that goes beyond the harmonic form represented from yearly temperatures. Within the Conclusion and Outlook section we propose future MESMER-M developments (e.g. of a module to represent land-cover effects) and emphasise the need that representation of other

forcings should be sufficiently decoupled from the GHG induced temperature response. We propose prefacing this part with the limitations of MESMER-M's applicability i.e. it will fall short when the overall mean response in monthly temperatures is dominated by other forcings that are decoupled from the GHG induced temperature response.

Specific comments

(1)In the Abstract, it should be 'model projection uncertainty' instead of 'model uncertainty. Model uncertainty comes from incomplete representation of physical processes, uncertain/unknown physical parameterisations, structural uncertainty.

 We will change "model uncertainty" to "model projection uncertainty"

(2) Line 5, it should be 'selected climate variables' instead of 'select climate variables'.

 This change will be implemented accordingly

(3) Line 10, what does 'mean response' refer to here?  Also this is an odd sentence structure, consider reframe to 'represent the monthly temperature cycle in response to the yearly temperatures'.

 Mean response refers to the direct response to evolving yearly temperatures, we will rephrase to:

'represent the mean monthly temperature response to yearly temperatures'

(4) Line 50 & in the Abstract, it's important to be clear which one the authors are trying to refer to, internal variability or natural variability, to me these are different things.

 We will edit such that we consistently use natural variability

(5) Line 64, is there any particular reason why a spatial resolution of 2.5 by 2.5 is used? Could the authors comment on how they expect the results to change if the analysis is done at a lower or higher resolution, and does the training of the emulator have any requirement or restriction when it comes to spatial resolution?

 A 2.5° by 2.5° grid is used simply as it is the best compromise to getting all climate models to the same resolution, thus allowing comparison of emulator applicability between models. Unfortunately fully investigating the effect of spatial resolution is beyond the scope of this study, we wouldn't expect it to have much of an effect however apart from slowing down or speeding up the emulator training time.

(6) Line 66, please specify why this reference period is chosen.

 This was for consistency with the paper describing the MESMER emulator (Beusch et al. 2020). We will specify this within L66.

(7) Line 101, the authors need to explain Bayesian Information Criterion to the readers (the significance of BIC), and why 8? Figure 2 suggests you are using 6 instead of 8.

We will modify L101 to read as:

"... we quantify the balance between the model complexity and accuracy using the Bayesian Information Criterion…"

We performed the BIC till n=8, however n greater than 6 was never chosen. We thus show an upper limit of 6 in the top panel's colorbar for Figure 2. For sake of simplicity we can however change n = 6 in L101 also.

(8) Line 121, please explain the Yeo-Johnson transformation, for the benefit of the readers who are not familiar with this, which I expect would be the case for many readers.

We accept that further explanation is desirable and will add the equation for the Yeo-Johnson transformation before equation (5).

(9) Line 127, I don't understand how and what fitting is being done using maximum likelihood here?

The epsilon coefficients which the Yeo-Johnson's lambda parameter is defined from are being fitted for. Equation 5 defines lambda and we state that epsilon_{0,m,s} and epsilon_{0,m,s} are the coefficients fitted for using maximum likelihood.

(10) Line 134, again, for the benefit of the readers, please specify what a Gaspari-Cohn function is here and why you choose this function to apply here.

The Gaspari-Cohn function was used in previous MESMER developments (see Beusch et al. 2020, equation 8) and allows for exponentially vanishing correlations with distance such that anisotropy of spatial cross-correlations on regional scales is still retained. We chose it for consistency within MESMER as well as the aforementioned property. Since it was already elaborated on in the previous MESMER paper we do not go into its specifics. To make this clearer however we can add to L134

"... localized by point-wise multiplication with the smooth Gaspari Cohn correlation function (Gaspari and Cohn, 1999) which has exponentially vanishing correlations with distance r_m."

(11) Line 156, I don't understand the authors' decision to only look at the top 50 highest power spectra. Please elaborate your thinking and reasoning behind this.

We initially wanted to check how well the emulated power spectra corresponded to that of the ESM runs by looking at how well we represent the frequencies where power within the signal is most concentrated (thus focussing on the 50 highest power spectra). From insight of reviewer 1 however, we will change this to look at how well we represent the spectra of the highest frequencies that occur within the ESM runs (see comment 10 of reviewer 1 for more details).

(12) Line 166-168, please consider rewriting this bit to clarify what's exactly being done to create these emulated quantiles. This part reads very confusing as it is now. In the following sentences, the quantile comparison description also lacks clarity.

This was also identified by referee 2 and we propose providing a numbered step-by-step procedure of this within the section.

(13) Section 3.4. Please explain why these particular biophysical variables (as listed in Table 1) are considered (chosen over other variables) and used in this study.

We mainly wanted a representation on changes in radiative and thermal fluxes for which these variables provide the most information.

(14) Line 195, I don't quite understand how this procedure is done, how did the authors use the physical model to augment the harmonic model results. Please elaborate.

By "augment" we simply mean that we add the phyiscal model predicted variability to the harmonic model results.

(15) Line 224-225, the authors should consider adding some discussions here on why these two models show such outlier behaviour.

Unfortunately we could not think of any obvious reason for such model behaviour. We additionally refrained from going into too detailed analysis of each model as that would require a more in-depth analysis of the model formulation itself, which is beyond the scope of this study.

(16) Section 4.2, please explain why these 4 ESMs are presented here, how they are representative (e.g. span across some projection range), and why WNA and WAF regions are chosen here.

We will add that these models represent divers genealogies according to \cite{Knutti2013} and \cite{Brunner2020}.

(17) Figure 3 & Figure 4. The labelling on the upper left corner should be 4 ESMs.

In this case we mean the number of ESM runs we are showing and not ESMs. We indeed show 4 different ESMs but for each ESM 3 runs are plotted.

(18) I would suggest changing the whisker colors in Figure 6 & 7 so that it's easier to see them.

This is a good suggestion and we will try darken the whiskers

(19) Table A1 clearly shows the split between training/test runs is not always 70/30, as opposed to what's stated in the text. Please check and confirm what's being done

We roughly followed a 70-30 split, however for MIROC6 and CanESM5 a 50-50 split was done, as training on more than 10 ensemble members led to significant training time with no real gain in model performance. We can modify L61-62 to read as:

"...is done in a roughly 70-30 manner, and for models with more than 20 ensemble members a 50-50 manner so as to maintain a good balance between training time and model performance."

---

## Author Response (AR2)

Shruti Nath
Climate Analytics
Ritterstrasse 3, 10969, Berlin, Germany
Phone: +49 (0) 30 259 22 95 44
E-mail: shruti.nath@climateanalytics.org

[Figure]

Earth System Dynamics editorial board Berlin,
18 March 2022

**MESMER-M: an Earth System Model emulator for spatially resolved monthly
temperature**

Dear Dr. Messori,

Please find enclosed the manuscript after implementing the minor revisions from the second
round of revisions as well as a version with tracked changes. Point by point answers to the
reviews are directly attached with this letter. Most of the changes are small textual
edits/clarifications and corrections of grammatical errors and captions (arising from both the
reviews and a round of proof reading).

We think that the minor reviews have been extremely helpful in strengthening the quality of
the paper and are grateful to all those involved in the review process.

Sincerely,

Shruti Nath

(on behalf of all the co-authors)

**Reviewer 1**

Thank you to the authors for your patience in my somewhat delayed review and for the detailed responses to the questions I raised in my initial review. I believe that the authors have done a reasonable job of responding to the questions I raised and I would be happy to see this article published in more or less its current form. The following are a few comments that might be worth considering, but I believe the authors should have the final say about whether or how best to address items 1-2 below.

We once again thank the reviewer for their input, it has vastly enriched the technical quality of the paper. We hope to have addressed the remaining points listed below sufficiently.

1. While I appreciate that you took my suggestion to also pay attention to the spectrum at high frequencies, I do not understand the choice to switch to looking at only the 50 highest frequencies rather than the 50 lowest frequencies, instead of simply looking at the agreement of the spectra over all frequencies (as was my intended suggestion). I do not think this is such a big deal, but the choice here seems somewhat arbitrary and the low frequencies do matter too.

We recognise that checking for the whole range of frequencies is important, but after further consideration we decided to stay focussed on the higher frequencies since this is also what we are most interested in capturing with the AR(1) process.

2. Choosing the order of the seasonal model to minimize the AIC using a likelihood that assumes temporal independence will, I believe, tend to result in including too many terms in the seasonal model. The reason is essentially that the likelihood assuming independence thinks that the effective sample size is larger than it actually is. Again I don't think this is such a big deal -- and I think it's better to include a few too many terms in the seasonal model than too few -- but just as a point of good statistical practice it might be worth briefly acknowledging this fact.

A line (L110) acknowledging this shortcoming of using the AIC was added

3. Overall I do think that the article could use a round of proofreading. I'm sorry that I don't have the time to point out specific grammatical and spelling issues, but there are a number of them so I'd recommend attending to this.

Proof reading was done and we hope all grammatical errors were addressed.

**Reviewer 2**

I understand and accept the authors' responses to my previous concerns. The draft appears to need minor revisions for clarification. The followings are further comments to be considered including those for editorial suggestion.

We thank the reviewer for their further comments, they definitely improve the quality of the text and its readability. We have implemented most of the suggestions as elaborated below.

L8. It would be informative to notify that the MESMER is a statistical emulator, which is distinct from a physical climate emulator like MAGICC. This is also the case for Introduction.

For ease of readability we do not elaborate on this in the abstract but make it quite clear in the Introduction (L32)

L13-14. Briefly describe how the performance is verified.

We have added this to the line

L93. 'other' should be capitalized.

This has been done

L103. I notice some minor differences between this equation and the one displayed in Figure 1, which should be harmonized if appropriate.

We have corrected this

L106. beta_1 = 1 (not beta_0)

Has been corrected

L117. Throughout the text and figure captions, 'Gaussian' and 'gaussian' are mixed.

We have changed all to Gaussian

L122. This sentence should not be indented.

Indent removed

L130. Here, 'power transformed' may be a bit confusing. It would be good to explicitly mention that Equation (3) is a type of power transformation.

We have changed this to "normalised as according to equation 3."

L151. The term 'innovations' is a bit confusing. It would be good to rephrase it if possible.

We have changed this to "noise terms"

L192. I notice several sentences including a one-digit number like here. The ESD style guide specifies to use words for cardinal numbers less than 10 for items other than units of time or measure.

We have changed this

L194. I do not think 'pinpoint' is an appropriate wording.

Changed to identify

L197. Is 'physical approach' in the subsection title a suitable wording? The GBR model uses physical variables, but the method itself is a statistical approach.

Changed to physically-informed approach

L200-201. ENSO is a variability mode associated with atmosphere-ocean interaction. It is not suitable for an example of atmospheric processes although resulting global teleconnections occur through atmospheric processes.

We have made this clearer

L245. 'shift' seems vague.

Replaced by "has a relatively low amplitude"

L253. Delete 'lambda'.

Deleted

L241. What is pointed by 'aforementioned' seems vague.

Replaced by "The source of high equatorial $\widetilde{\lambda}_{m,s}$ values "

L266. This sentence should start with 'For July' for clarification.

Sentence has been changed to "July shows similar behaviour for $\gamma_{1,m,s}$ across most ESMs, albeit with a larger spread in values
"

L278. 'by the mean response module and the residual variability module' (maybe 'and' is missing)

And was added

L288. What is described by 'followed by stabilisation' is vague.

Replaced by "non-linear"

L312. 'between the ESM training runs and emulations for a given ESM' (maybe 'and' is missing)

And added

L323-325. It needs to be explained why the value remains high even as the distance increases for the test runs.

"where spatial cross-correlations are not localised" was added

L373-375. Regarding the results from the benchmarking approach shown in Figure 13, it is hard to identify the description about 'with only MIROC6 and MPI-ESM1-LR showing larger distances for the full emulator in the Indo-Gangetic region, South America and Central-West Africa.'

Modified to "MIROC6 and MPI-ESM1-LR showing larger distances for the full emulator in the Indo-Gangetic region and Central-West Africa respectively"

L393-394. ENSO itself is not a suitable example for regional-scale changes. Regional-scale changes occur through the modulation of global-scale atmospheric circulation patterns.

Elaborated upon by adding " modulation of atmospheric circulation patterns due to changes in ENSO"

L404-405. Given the results described in 5.4, the authors' view as mentioned 'in most cases even better than the physically-based model' is a bit questionable.

We believe that from figure E2 this statement holds (of course with the limitations identified in section 5.4)

L435-443. I suppose that warming-induced changes in major climate modes and their ESM dependencies have been relatively well documented in literature in terms of specific indexes, which may lead to improved emulation of local temperature variability associated with such modal shifts. I will not ask for further revision on this issue, but I leave this comment for clarification.

We changed this to "potentially coupled"

Figure 1. Greatly improved.
As the green panels illustrating the inverse Yeo-Johnson transformation are rather complex, a brief description would be helpful.
In the global maps, I prefer to using a common color scale for the harmonic component and the total, but no problem if the authors' choice is better for some reason.
Period is missing after 'E.g' in the figure text 'E.g Inverse Yeo-Johnson power transformation.'

Periods in the E.G. text are elaborated on and a better caption description given

Figure 2. Consider to use mathematical symbols gamma_{1,m,s} and r_m at 'the local lag-1 autocorrelation coefficients' and at 'the localisation radii' in the caption.

Has been applied

Figure 8-11. For consistency, the number of the training runs should be indicated as for the test runs.

The label 'Test runs' between the top and bottom blocks is not needed.
In the caption, '(colour)', not '(colour(.'

Typo corrected. For consistency with other figures where only test runs are shown we decided to keep only the test run numbers.

Figure B1 and C1. Describe the percentage numbers attached to each panel.

Description provided

Figure B2. Same as Figure B1, not Figure C1.

Figure C2. Same as Figure C1, not Figure D1.

Captions corrected

References. It seems that two discussion papers Beusch et al. and Brunner et al. have already been published or accepted.

Has been corrected

**Reviewer 3**

The authors did a great job to answer reviewer's comments. I am especially pleased with the added discussions to the Conclusion and Outlook section, which now states the limitations of the current approach and proposes direction for further improvements.

We once again express gratitude to the reviewers comments, they have vastly helped the text and improved the outlook.

---

## Author Response (AR3)

Shruti Nath
Climate Analytics
Ritterstrasse 3, 10969, Berlin, Germany
Phone: +49 (0) 30 259 22 95 44
E-mail: shruti.nath@climateanalytics.org

[Figure]

Earth System Dynamics editorial board
Berlin, 29 March 2022

**MESMER-M: an Earth System Model emulator for spatially resolved monthly temperature**

Dear Dr. Messori,

Attached is the final manuscript with the editorial corrections implemented (see L172). We are delighted to be able to introduce MESMER-M through the Earth System Dynamics journal.

Sincerely,

Shruti Nath

(on behalf of all co-authors)